# Reliable Fine-Grained Evaluation of Natural Language Math Proofs

**Wenjie Ma**[1*]   **Andrei Cojocaru**[1†]   **Neel Kolhe**[1†]   **Haihan Zhang**[3†]
**Vincent Zhuang**[2‡]   **Matei Zaharia**[1]   **Sewon Min**[1,4]
[1] UC Berkeley   [2] Google DeepMind   [3] Peking University   [4] Allen Institute for AI

## Abstract

Recent advances in large language models (LLMs) for mathematical reasoning have largely focused on tasks with easily verifiable final answers while generating and verifying natural language math proofs remain an open challenge. We identify the absence of a reliable, fine-grained evaluator for LLM-generated math proofs as a critical gap. To address this, we propose a systematic methodology for developing and validating evaluators that assign fine-grained scores on a 0–7 scale to model-generated math proofs. To enable this study, we introduce **Proof-Bench**, the first expert-annotated dataset of fine-grained proof ratings, spanning 145 problems from six major math competitions (USAMO, IMO, Putnam, etc.) and 435 LLM-generated solutions from Gemini-2.5-Pro, o3, and DeepSeek-R1. Using PROOFBENCH as a testbed, we systematically explore the evaluator design space across key axes: the backbone model, input context, instructions and evaluation workflow. Our analysis delivers **ProofGrader**, an evaluator that combines a strong reasoning backbone LM, rich context from reference solutions and marking schemes, and a simple ensembling method; it achieves a low Mean Absolute Error (MAE) of 0.926 against expert scores, significantly outperforming naive baselines. Finally, we demonstrate its practical utility in a best-of-$n$ selection task: at $n = 16$, PROOFGRADER achieves an average score of 4.14/7, closing 78% of the gap between a naive binary evaluator (2.48) and the human oracle (4.62), highlighting its potential to advance downstream proof generation.

## 1 Introduction

| | |
|---|---|
| **USAMO 2025 P1.** Fix positive integers $k$ and $d$. *Prove* that for all sufficiently large odd positive integers $n$, the digits of the base-$2n$ representation of $n^k$ are all greater than $d$. | **IMO 2024 P2.** Determine all positive integers $a$ and $b$ such that there exists a positive integer $g$ with $\gcd(a^n + b, b^n + a) = g$ for all sufficiently large $n$. |

Figure 1: Two example proof problems selected from well-established competitions.

Large language models (LLMs) have recently achieved remarkable progress in mathematical reasoning, attaining strong performance on a variety of benchmarks. Such models are especially strong at solving final-answer problems because they can be trained using reinforcement learning against simple answer verifiable rewards (Shao et al., 2024; DeepSeek-AI et al., 2025; Yang et al., 2025; Yu et al., 2025; Wang et al., 2025). However, these methods do not transfer to proof generation for two reasons: (i) many proof problems do not admit a single, easily checkable final answer (Figure 1 left); and (ii) even when a final answer exists (Figure 1 right), verifying it is insufficient to assess proof validity, as the reasoning may contain substantial intermediate errors (Petrov et al., 2025; Dekoninck et al., 2025). Because proof-generation tasks constitute a large share of mathematical problem solving in research and education, this necessitates reliable proof evaluation methods.

We identify **the absence of a reliable proof evaluator** as a key bottleneck for improving proof generation, which is essential for providing faithful assessments of model capabilities and accurate reward signals for training models. Expert grading, while accurate, is slow and costly. While formal math (e.g., Lean) offers absolute certainty, it remains detached from the natural language used in

---

*Correspondence: `windsey@berkeley.edu`

†Dataset contributors.

‡Served in an advisory capacity only.

most human mathematics education and research; furthermore, automatically translating natural-language proofs into formal languages is brittle and remains extremely challenging (Gao et al., 2025; Liu et al., 2025a). Our work therefore focuses on the critical and complementary task of evaluating proofs in their natural representation. While LLM-as-a-judge (Zheng et al., 2023a; Sheng et al., 2025; Dekoninck et al., 2025; Huang & Yang, 2025) is promising, its application to math proofs is unsettled, and outcomes are sensitive to evaluator design—model choice, available context, rubric construction, and prompting—all of which are poorly understood.

This paper tackles this bottleneck by developing a methodology to create high-fidelity, fine-grained evaluators, and demonstrates for the first time that they can nearly match human oracle performance in downstream tasks. Our objective is to design an automated evaluator, $\mathcal{E}$, that takes a problem $p$, a model-generated solution $s$, and an optional set of contextual materials $\mathcal{C}_{\text{context}}$ to produce a fine-grained integer score, $\hat{y} \in \{0, 1, \ldots, 7\}$. We identify the optimal evaluator not through model training, but by *searching over a discrete space of configurations* $\mathbb{C}$. We adopt the 0–7 scoring scale used in premier competitions to enable nuanced assessment beyond binary correctness.

First, to support our study, we introduce **PROOFBENCH**, the first expert-annotated dataset for fine-grained proof evaluation that spans problems from multiple contests and years. It spans 145 problems from EGMO, USAMO, IMO, USA TST, APMO, and PUTNAM (2022–2025), with 435 LLM-generated solutions from state-of-the-art models (GEMINI-2.5-PRO, O3, DEEPSEEK-R1). Data annotation follows a two-stage process: (i) generate a problem-specific marking scheme to standardize criteria while allowing valid alternative approaches; (ii) have experts score each solution with that scheme while allowing for valid alternative solutions.

Using PROOFBENCH as a testbed, we systematically explore the evaluator design space, including four primary axes: backbone models, input context (such as reference solutions and problem-specific marking schemes), instruction sets and evaluation workflows. We consider single-pass evaluators as well as more advanced techniques, including (1) ensembling multiple evaluation runs and (2) staged workflows that decompose the complex evaluation task into focused, sequential steps. Our analysis delivers **PROOFGRADER**, an LLM-based evaluator that combines a strong backbone LM with informative context (both reference solutions and a marking scheme) and simple ensembling that is surprisingly effective; it achieves a low Mean Absolute Error (MAE) of 0.926 against expert scores, significantly outperforming naive baselines.

Finally, we validate the evaluators' practical utility in a downstream best-of-$n$ selection task, a standard proxy for assessing its potential as a reward signal (Gao et al., 2022; Frick et al., 2024; Malik et al., 2025). At $n = 16$, PROOFGRADER achieves an average score of 4.14/7, closing 78% of the performance gap between a naive binary evaluator (2.48) and the human oracle (4.62). It also outperforms computationally intensive, pairwise selection methods such as Knockout tournament selection (Liu et al., 2025b). These results highlight PROOFGRADER's effectiveness in identifying high-quality proofs. To summarize, our contributions are three-fold:

- We introduce PROOFBENCH[1], an expert-graded math proof dataset consisting of problems from well-established contests and solutions from state-of-the-art reasoning models (§2).
- We conduct a systematic study of key evaluator design factors and introduce PROOFGRADER, our best-performing evaluator that combines a strong reasoning LM, a problem-specific marking scheme, and simple ensembling, achieving a strong alignment with expert ratings (§3).
- We show that PROOFGRADER is on par with experts in a best-of-$n$ selection task, improving a score from 2.48 (binary evaluator) to 4.14, highlighting its promise as a reward model (§4).

## 2 PROOFBENCH: EXPERT-RATED MATH PROOF SOLUTIONS

We assess proof evaluators for *human alignment* on a 0–7 scale, which requires fine-grained expert annotations on model-generated proofs across diverse sources and years. Existing resources are limited in scale or scoring granularity (Petrov et al., 2025; Dekoninck et al., 2025). We therefore construct our own dataset, PROOFBENCH, consisting of 145 proof problems from six major math competitions across four years, with model-generated solutions by the state-of-the-art reasoning models: o3, Gemini-2.5-Pro, and DeepSeek-R1-0528. This section describes the data collection, annotation process, and statistics; §3 then describes how we use it to evaluate proof evaluators.

---

[1]The full dataset is available at `huggingface.co/datasets/wenjiema02/ProofBench`.

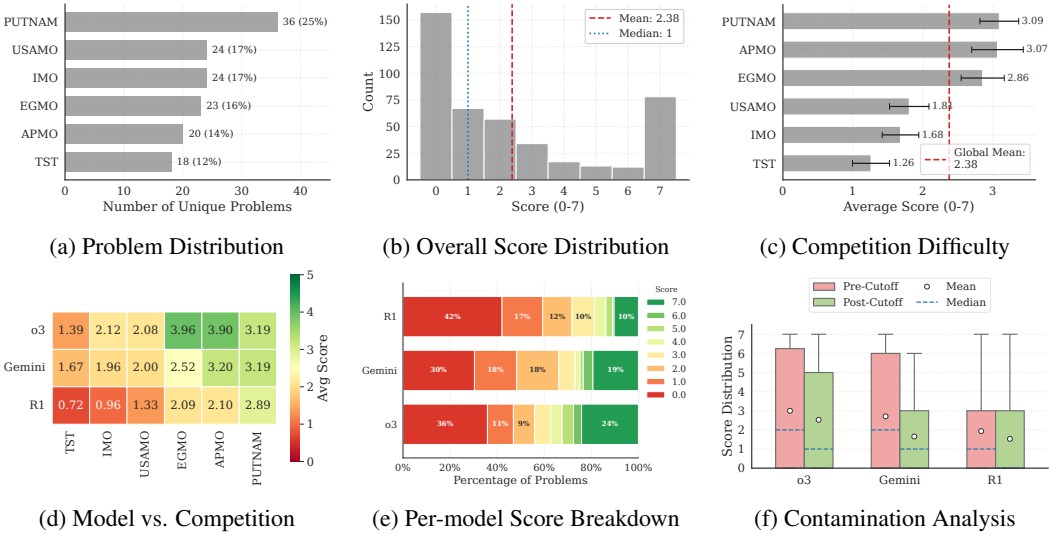

Figure 2: **Dataset information.** (a) Distribution of unique problems across the six competitions. (b) Aggregate histogram of scores (0–7) across all models. (c) Competition difficulty ranking based on average scores; Putnam yielded the highest performance (3.09), while TST was the most challenging (1.26). (d) Heatmap of average scores by model and competition (Green indicates higher performance). (e) Stacked bar chart detailing the percentage of problems achieving each score (Red=0, Green=7). (f) Per-model box plots comparing performance on problems released before vs. after the model's knowledge cutoff. There is a consistent drop in mean and median scores post-cutoff.

**Why 0–7 Scale?** The 0–7 scale aligns the dataset with the established grading standards of premier mathematics competitions[2], the source of our benchmark problems. Moreover, its fine-grained nature is critical for a nuanced assessment of proof quality, which is not captured by the binary "correct"/"incorrect" grading, as we show throughout §3 and §4.

## 2.1 DATASET CONSTRUCTION

**Problem Sources.** We collected 145 problems from the official websites of prestigious mathematics competitions, including the APMO, EGMO, IMO, Putnam, USA TST, and USAMO, spanning the years 2022–2025. Whenever possible, problems were parsed from official PDFs, normalized, and accompanied by all available human-written solutions. This ensures the reliability of both problem statements and ground-truth solutions, avoiding transcription errors commonly found in secondary sources. When official solutions were unavailable, we supplemented the data with materials from reliable third-party websites. More details about each competition are provided in §B.

**Proof Generation.** For each problem, we generate a proof from models (*generators*) using a standardized prompt (§K) that asks for a complete, self-contained proof. We generate proofs from three state-of-the-art reasoning models: OpenAI o3 (OpenAI, 2025), Gemini-2.5-Pro (Google DeepMind, 2025), and DeepSeek-R1-0528 (DeepSeek-AI et al., 2025), which span both proprietary and open-source families.

**Expert Grading Pipeline.** Finally, annotation proceeds in two stages: marking scheme generation and model-generated proof grading. All annotations were conducted by a team of five experts with Putnam-level or national Math Olympiad experience, using a carefully designed web interface.

**Stage 1: Marking Scheme Finalization.** A central challenge in creating this benchmark was ensuring consistent and scalable expert grading. While human-written rubrics are the gold standard, automated generation offers superior scalability and a systematic way to recognize key steps from provided reference solutions. In our method, the marking scheme is produced by an LLM $\mathcal{M}_{\text{MS}}$.

---

[2]The official Putnam Competition uses a 0–10 scale. For consistency across competitions, we normalize all expert annotations to a unified 0–7 scale.

Given the problem $x$ and reference solutions $\mathcal{S}$, $\mathcal{M}_{\mathrm{MS}}$ is prompted to output (i) a list of conditions under which scores are awarded or deducted, and (ii) a list of trivial cases that should not be awarded any points. This generator was developed through a rigorous refinement process where experts also graded the quality of the generated marking schemes, providing feedback that led to the selection of gemini-2.5-pro as the final $\mathcal{M}_{\mathrm{MS}}$ (see §C for details). An example of a generated marking scheme together with problem text and reference solution is also provided in §C (Figure 4). From our expert evaluations, approximately 85% of the generated marking schemes in our final dataset were judged to be reasonable and of high quality.

**Stage 2: Proof Grading.** With a problem-specific marking scheme, an expert annotator scores a given model-generated proof. Experts were instructed to treat the marking scheme as a detailed reference for the expected solution path, rather than a rigid checklist. This is particularly important for fairly evaluating proofs that employed a novel method different from the provided ground-truth solutions, ensuring that valid alternative reasoning paths were credited appropriately. The experts will assign a score on a 0–7 scale. Before large-scale annotation, our experts underwent a calibration phase to align their judgments and finalize rules for handling edge cases, a critical step for minimizing inconsistencies. We assign 41% of the solutions to more than one annotator and their within-1-point agreement is 87.5%. Disagreements are further resolved through discussions.

## 2.2 DATASET STATISTICS

In total, PROOFBENCH consists of 145 proof problems and 435 expert-annotated evaluations of proofs generated by OpenAI o3, Gemini-2.5-Pro, and DeepSeek-R1-0528. Figure 2 summarizes the dataset statistics, including the problem distribution across competitions (Figure 2a) and the aggregate score frequencies (Figure 2b).

Our analysis reveals several key findings about state-of-the-art reasoning models:

- As shown in Figure 2e, current models remain far from reliable; even the strongest models achieve scores of 6 or higher on fewer than 30% of problems. OpenAI o3 demonstrates the strongest overall performance with the highest mean scores in 5/6 competitions (Figure 2d).
- Performance varies considerably across problem sources (Figure 2c, Figure 2d). Models perform best on PUTNAM (mean 3.09) but struggle markedly on TST (mean 1.26).
- Comparison of performance on problems released before versus after the knowledge cutoff (Figure 2f) reveals a consistent drop in scores on newer problems, suggesting potential contamination in the training data or a generalization gap on novel problems.

## 3 A SYSTEMATIC STUDY OF EVALUATOR DESIGNS

Using PROOFBENCH, we systematically study the key design factors for a math proof evaluator that produces fine-grained scores (0–7) for model-generated solutions. We demonstrate that (i) a strong reasoning backbone with informative context (like marking schemes) yields substantial gains, and (ii) simple ensembling further improves accuracy and robustness.

### 3.1 EVALUATOR DESIGNS

Given the task of scoring model-generated solutions, a naive evaluator would simply take a problem and a generic instruction and assign a score. Instead, we systematically investigate design choices that enhance evaluation quality beyond this baseline. We first analyze single-pass evaluators along three key dimensions: (i) backbone model, (ii) contextual input, and (iii) instruction set. We then extend this design to ensemble-based and multi-stage evaluation workflows.

**Single-Pass Methods.** A single-pass evaluator prompts a backbone model $\mathcal{M}$ (which may or may not be the same as $\mathcal{M}_{\mathrm{MS}}$) to grade a solution $s$ in one step. We analyze single-pass evaluators along three dimensions: the backbone LM, the context provided and the instructions given.

- **Backbone Model Choice.** This refers to the LLM that is prompted to execute the evaluation. We compare six models that span different model families, sizes and reasoning capabilities: O3, GPT-5 (GPT-5-Thinking (OpenAI, 2025)), GEMINI (Gemini-2.5-Pro), O4-MINI (OpenAI, 2025), R1 (DeepSeek-R1-0528), and GPT-4O (OpenAI, 2024).
- **Context.** We consider four different context configurations, including: providing both the pre-generated marking scheme and the reference solution(s) (REF+MS); providing only the marking

| Model | Context | RMSE ↓ | MAE ↓ | WTA$_{\leq 1}$ (%) ↑ | Kendall-$\tau$ ↑ | Bias ≈ | Quality |
|---|---|---|---|---|---|---|---|
| **O3** | REF+MS | **1.273** ± 0.16 | **0.964** ± 0.12 | **76.5** ± 4.3 | **0.502** | **-0.008** | ■■■ |
| | MS | 1.418 ± 0.17 | 1.069 ± 0.14 | 72.8 ± 4.3 | 0.477 | -0.381 | ■■■ |
| | REF | 1.575 ± 0.14 | 1.330 ± 0.12 | 65.3 ± 4.6 | 0.481 | 0.478 | ■■■ |
| | NONE | 1.901 ± 0.15 | 1.680 ± 0.15 | 49.5 ± 5.8 | 0.435 | 0.924 | ■■■ |
| **GEMINI** | REF+MS | 1.696 ± 0.17 | 1.342 ± 0.15 | 62.7 ± 4.9 | **0.529** | 0.626 | ■■■ |
| | MS | **1.502** ± 0.17 | **1.142** ± 0.14 | **70.0** ± 4.2 | 0.488 | **0.151** | ■■■ |
| | REF | 2.177 ± 0.14 | 1.910 ± 0.15 | 39.9 ± 5.2 | 0.410 | 1.285 | ■■■ |
| | NONE | 2.397 ± 0.15 | 2.107 ± 0.16 | 36.6 ± 4.9 | 0.319 | 1.496 | ■■■ |
| **O4-MINI** | REF+MS | 1.816 ± 0.22 | 1.367 ± 0.19 | 67.6 ± 4.6 | 0.476 | 0.762 | ■■■ |
| | MS | **1.636** ± 0.22 | **1.234** ± 0.18 | **69.5** ± 4.8 | **0.505** | **0.309** | ■■■ |
| | REF | 1.858 ± 0.19 | 1.504 ± 0.16 | 61.3 ± 4.8 | 0.465 | 0.950 | ■■■ |
| | NONE | 2.276 ± 0.23 | 1.914 ± 0.21 | 49.5 ± 5.3 | 0.430 | 1.569 | ■■■ |
| **GPT-5** | REF+MS | 1.353 ± 0.16 | 1.055 ± 0.13 | 73.2 ± 4.6 | 0.532 | 0.295 | ■■■ |
| | MS | **1.350** ± 0.17 | **1.018** ± 0.14 | **74.9** ± 4.2 | **0.536** | **-0.175** | ■■■ |
| | REF | 1.617 ± 0.13 | 1.400 ± 0.12 | 57.0 ± 5.4 | 0.501 | 0.799 | ■■■ |
| | NONE | 1.919 ± 0.14 | 1.708 ± 0.14 | 46.5 ± 5.4 | 0.404 | 1.034 | ■■■ |
| **R1** | REF+MS | 1.735 ± 0.22 | 1.357 ± 0.18 | 66.4 ± 5.2 | 0.429 | 0.732 | ■■■ |
| | MS | **1.682** ± 0.22 | **1.298** ± 0.18 | **68.5** ± 4.9 | **0.450** | **0.422** | ■■■ |
| | REF | 3.187 ± 0.23 | 2.736 ± 0.22 | 30.3 ± 5.0 | 0.289 | 2.450 | ■■■ |
| | NONE | 3.273 ± 0.25 | 2.842 ± 0.25 | 33.5 ± 4.9 | 0.102 | 2.581 | ■■■ |
| **GPT-4O** | REF+MS | 2.599 ± 0.20 | 2.197 ± 0.20 | 39.7 ± 5.0 | **0.479** | 1.824 | ■■■ |
| | MS | **2.245** ± 0.21 | **1.827** ± 0.19 | **50.4** ± 5.3 | 0.377 | **1.001** | ■■■ |
| | REF | 2.726 ± 0.19 | 2.371 ± 0.18 | 36.0 ± 4.9 | 0.343 | 1.887 | ■■■ |
| | NONE | 3.402 ± 0.26 | 3.001 ± 0.26 | 31.9 ± 4.9 | 0.208 | 2.614 | ■■■ |

Table 1: **Performance comparison of 6 LLMs on 0-7 scale evaluation tasks under different context designs.** Contexts: REF+MS (reference + marking scheme), MS (marking scheme only), REF (reference only), NONE (no context). Values shown as mean with ± 95% CI. Best values per model in **bold**; best (■) and worst (■) context highlighted. Arrows: ↓ lower better, ↑ higher better, ≈ closer to zero better. Quality: ■ Excellent, ■ Good, ■ Fair, ■ Poor, ■ Bad.

scheme (MS); providing only the reference solution(s) (REF); and a naive baseline where only basic grading instructions without any problem-specific context are provided (NONE).

- **Instruction.** The instruction set refers to the prompt that guides the evaluator on how to interpret and apply the provided context and perform the task. In our study, for the most informative context setting, REF+MS, we compared 3 types of instructions to understand how guidance on using the context affects performance. These include NORM (Normal), a flexible instruction that directs the model to follow the marking scheme but allows it to map valid alternative approaches to equal checkpoints; STRICT requires the model to adhere strictly to the provided marking scheme and penalize any deviation; and BASIC has minimal guidance on how to use the provided materials.

**Ensemble.** We consider a simple ensembling technique, which runs the same evaluator independently multiple times and combines the individual ratings with an aggregation operator, such as the mean or median. This technique is expected to produce a more stable final score, reducing the variance of single-pass evaluators.

**Staged Evaluation.** We consider staged workflows that decompose the complex task of evaluation into a sequence of more focused reasoning steps. We implement **Binary & Errors → Fine-Grained**, which first predicts the binary correctness of the proof and identifies key errors, and then makes a second pass that uses this output to generate a more calibrated, 0–7 score.

## 3.2 EVALUATION METRICS

We evaluate on the 0–7 scale using per-problem metrics. All problems have the same number of responses ($n$). Let $P$ be the number of problems. For each problem $p = 1, \ldots, P$ with expert scores $y_{pi}$ and evaluator outputs $\hat{y}_{pi}$, we compute

$$\text{MAE}_p = \frac{1}{n} \sum_{i=1}^{n} |\hat{y}_{pi} - y_{pi}|, \quad \text{RMSE}_p = \sqrt{\frac{1}{n} \sum_{i=1}^{n} (\hat{y}_{pi} - y_{pi})^2},$$

$$\text{Bias}_p = \frac{1}{n}\sum_{i=1}^{n}(\hat{y}_{pi} - y_{pi}), \quad \text{WTA}_p(\leq 1) = \frac{1}{n}\sum_{i=1}^{n}\mathbf{1}\{|\hat{y}_{pi} - y_{pi}| \leq 1\}.$$

MAE and RMSE both measure errors relative to expert scores (lower is better) but RMSE penalizes large mistakes more. $\text{WTA}_{\leq 1}$ measures the fraction of predictions that land within one point of the expert score. Bias measures the average signed error (systematic shift), with positive values indicating over-scoring and negative values indicating under-scoring. For ranking agreement within a problem, we use Kendall's $\tau_b$ (ties-adjusted). Detailed definitions of Kendall's $\tau_b$ as well as aggregation formulas are provided in §E.

## 3.3 WHAT FACTORS IMPROVE SINGLE-PASS EVALUATOR?

**Effects of backbone model and contextual information.** Table 1 shows the results of single-pass evaluators across different backbone models and context settings. First, *the strength of the backbone model* strongly correlates with performance: moving from weaker to stronger models brings better calibrations (O3 leads in nearly all metrics). Second, *contextual information* consistently improves all metrics for every backbone, with the marking scheme (MS) contributing the majority of the gain relative to the reference solution alone (REF); combining both (REF+MS) provides a small additional improvement primarily for the strongest backbone (O3).

| Model | Instruction | RMSE ↓ | MAE ↓ | WTA$_{\leq 1}$ (%) ↑ | Kendall-$\tau$ ↑ | Bias ≈ |
|---|---|---|---|---|---|---|
| | NORM | **1.273** | **0.964** | **76.5** | **0.502** | **−0.008** |
| O3 | STRICT | 1.420 | 1.095 | 72.8 | 0.457 | −0.304 |
| | BASIC | 1.348 | 1.039 | 73.0 | 0.501 | 0.165 |
| | NORM | 1.816 | 1.367 | 67.6 | **0.476** | 0.762 |
| O4-MINI | STRICT | **1.718** | **1.266** | **69.2** | 0.443 | **0.396** |
| | BASIC | 1.817 | 1.360 | 67.6 | 0.437 | 0.724 |
| | NORM | 1.696 | 1.342 | 62.7 | **0.529** | 0.626 |
| GEMINI | STRICT | **1.581** | **1.231** | **67.4** | 0.475 | **0.316** |
| | BASIC | 1.773 | 1.424 | 61.7 | 0.469 | 0.832 |

Table 2: **Instruction ablation under REF+MS.** Three instruction styles guide how the evaluator uses context: NORM (flexible use of the marking scheme, allowing valid alternatives), STRICT (literal adherence with penalties for deviations), and BASIC (minimal guidance). For *Bias*, the closer to 0, the better; for other metrics, ↓ means the lower the better while ↑ means the opposite.

**Instruction style under REF+MS should match the backbone.** With context fixed at REF+MS, instruction choice modulates the calibration-ranking trade-off (Table 2). The strongest backbone (O3) attains the best overall accuracy and near-zero bias with the more flexible NORM prompt, whereas mid-tier models (e.g., GEMINI, O4-MINI) benefit from the more prescriptive STRICT prompt. This pattern suggests that stronger models can reliably and flexibly apply the marking scheme (e.g., mapping alternative derivations to equivalent checkpoints), while mid-tier models require more prescriptive guidance to reduce over-crediting and variance.

**Per-generator results.** A natural question that can arise is the potential bias when evaluators assess outputs generated by their own models. To investigate this, we evaluate three single-pass evaluators, each using the same reference solutions and marking scheme, on per-generator splits, i.e., response sets grouped by their generators (R1, GEMINI, O3). The results in Table 3 are mixed. On one hand, the strongest evaluator remains consistent across all generators: the O3-based evaluator. On the other hand, each evalua-

| Evaluator | MAE ↓ | | | Bias ≈ | | |
|---|---|---|---|---|---|---|
| | O3 | GEM. | R1 | O3 | GEM. | R1 |
| **O3** | **1.12** | **0.78** | **0.99** | -0.37 | 0.31 | -0.07 |
| GEMINI | 1.23 | 1.62 | 1.18 | -0.09 | 1.42 | 0.45 |
| R1 | 1.18 | 1.43 | 1.50 | 0.23 | 0.96 | 0.99 |

Table 3: **Per-generator evaluation accuracy.** Red cells indicate highest MAE and highest positive bias for each evaluator.

tor exhibits its highest MAE on outputs produced by its own model family, indicating a tendency toward within-generator underperformance. GEMINI and R1 exhibit a tendency to overscore their own responses, a bias not observed in the O3 evaluator.

## 3.4 Do Ensembling and Staged Workflows Improve Single-pass Evaluation?

We next extend single-pass evaluators through either ensembling or multi-stage pipelines.

| Model | Run | RMSE ↓ | MAE ↓ | WTA$_{\leq 1}$(%)↑ | kendall-$\tau$ ↑ | Bias ≈ |
|---|---|---|---|---|---|---|
| | Single (mean±std; n=5) | 1.265 (±0.039) | 0.981 (±0.028) | 75.5 (±1.126) | 0.509 (±0.022) | 0.018 (±0.020) |
| o3 | Best single | 1.225 | 0.944 | 77.1 | 0.540 | 0.009 |
| | MEAN | **1.169** | 0.940 | 69.7 | **0.578** | 0.018 |
| | MEDIAN | 1.185 | **0.926** | **77.7** | 0.540 | 0.008 |
| | MAJORITY | 1.186 | **0.926** | 75.6 | 0.523 | **0.004** |

Table 4: **Ensembling over multiple runs boosts performance and reduces variance for the o3 evaluator.** We compare five individual runs against three aggregation strategies. Both mean and median aggregation achieve a lower RMSE than the best single run. Mean aggregation is optimal for RMSE and ranking correlation (Kendall-$\tau$), while median aggregation excels on MAE and WTA$_{\leq 1}$.

**Ensembling helps.** For ensembling, we aggregate five independent o3 evaluation runs under REF+MS, with results reported in Table 4. Compared to the best single run, averaging all runs reduces RMSE by ∼0.06 (1.225→1.169) and improves rank agreement (Kendall-$\tau$ from 0.540 to 0.578). Median aggregation delivers the lowest MAE and highest WTA$_{\leq 1}$, and Majority matches that MAE while giving the smallest bias. Bolded cells in Table 4 mark these optima. Although absolute performance gain seems modest, one advantage of ensembling is reducing variance: single-pass evaluators, even with richer context, have high variance, whereas ensembling mitigates it.

| Model | Design | RMSE↓ | MAE↓ | WTA$_{\leq 1}$ (%)↑ | Kendall-$\tau$ ↑ | Bias≈ |
|---|---|---|---|---|---|---|
| o3 | **Single-pass (REF+MS)** | **1.273** | **0.964** | **76.5** | **0.502** | **−0.008** |
| | Binary+Errors → Fine-Grained | 1.375 | 1.065 | 73.0 | 0.444 | −0.102 |
| o4-MINI | Single-pass (REF+MS) | 1.816 | 1.367 | **67.6** | 0.476 | 0.762 |
| | **Binary+Errors → Fine-Grained** | **1.650** | **1.245** | 66.7 | **0.503** | **0.320** |
| R1 | Single-pass (REF+MS) | 1.735 | 1.357 | 66.4 | **0.429** | 0.732 |
| | **Binary+Errors → Fine-Grained** | **1.708** | **1.299** | **68.4** | 0.371 | **0.528** |
| o3+o4-MINI | Binary+Errors → Fine-Grained | 1.513 | 1.126 | 71.4 | 0.458 | −0.095 |

Table 5: Comparison of two-step **Binary+Errors → Fine-Grained** pipeline versus single-pass **(REF+MS)** evaluation across backbones. Staged design boosts mid-tier o4-MINI but degrades strong o3 performance; bold marks the best within each model block.

**Staged pipelines *may* improve weaker models, but not stronger ones.** For staged pipelines, Table 5 presents results for Two-step (Binary+Errors → Fine-Grained) workflow. We include o3-, o4-MINI- and R1-based evaluators. It improves performance on the o4-MINI and R1 backbone (e.g., reducing RMSE from 1.816 to 1.650 and MAE from 1.367 to 1.245 for o4-MINI) but hurts the strong o3 backbone (e.g., increasing RMSE from 1.273 to 1.375 and MAE from 0.964 to 1.065).

**Putting everything together: PROOFGRADER.** Based on our findings, we introduce PROOF-GRADER, the evaluator that is best-performing most consistently across all settings. PROOF-GRADER builds on the o3 model with the REF+MS configuration and the NORM instruction set, and incorporates simple ensembling techniques.

## 3.5 In-depth Analysis

**What are the common failure modes of PROOFGRADER?** To investigate failure cases, we manually inspected 50 solutions where 2 out of 5 ensemble runs deviated from human scores by at least 2 points. We find that over-crediting (10.8%) and under-crediting (12.2%) occur at similar rates. **Over-scoring** typically arises when PROOFGRADER relies on surface-level heuristics: it rewards solutions that mirror rubric structure or employ sophisticated frameworks (e.g., custom transformations) even when central claims are false. Furthermore, it often misclassifies fatal logical errors,

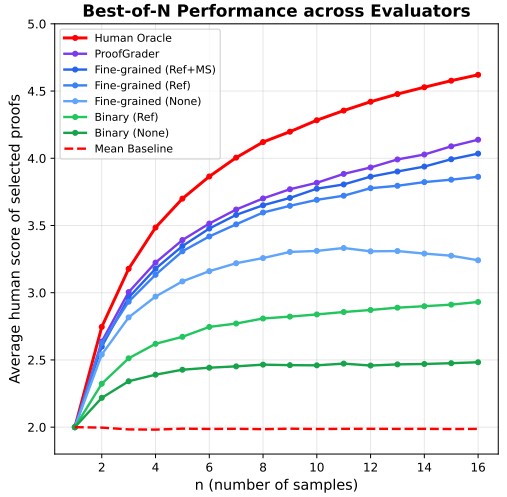
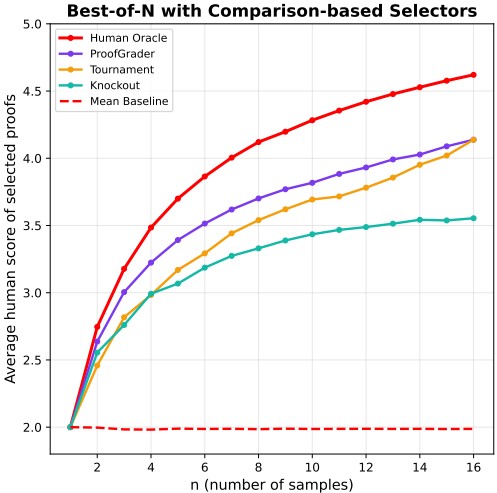

(a) **Evaluators as BoN selectors.** Ensemble-based fine-grained evaluator closely tracks the Human-Oracle curve, while the binary evaluators (green lines) perform much worse.

(b) **Comparison-based selectors.** Our fine-grained evaluator performs the best. Some strategies show signs of performance degradation at larger values of $n$, while our approach remains robust.

Figure 3: **Best-of-$n$ with different *selector*s.** Average best-of-$n$ score over 29 problems for O3 (generator) as $n$ increases from 1 to 16. PROOFGRADER is our ensemble evaluator: the median over five independent O3 runs using the reference solution and marking scheme.

such as broken reductions, as minor justification issues. **Under-scoring** commonly results from penalizing early missteps in otherwise correct proofs, double-counting a single flaw, or demanding excessive micro-justifications. By domain, failures are most frequent in Algebra and Geometry ($\sim$25% each). See §H for details.

**Why are reference solutions and marking schemes helpful?** Comparing evaluators based on O3 with NONE to those given a REF+MS, we find that in over 60% of cases, the evaluator without context *overestimates* the generation's correctness, i.e., it incorrectly assigns a *higher* score to the generation. Crucially, this overestimation is not uniform: we observe a strong correlation ($r = 0.699$, $p < 0.001$) between proof quality and evaluation gap. For *low-quality* proofs (scored 0–2), the no-context evaluator over-scores by an average of 1.7 points. In contrast, for *high-quality* proofs (scored 5–7), the evaluator with context information assigns *higher* scores by 0.8 points on average. This asymmetric pattern supports our hypothesis that, without reference solutions or marking schemes, the evaluator struggles to gauge proof *progress*.

**What if the evaluator uses a different marking scheme from human experts?** In our main setup, evaluators use the same marking schemes as the human graders. To test sensitivity to the rubric, we replace these with two alternatives: (i) regenerate a marking scheme using the *same* prompt/model (new sample), and (ii) generate a marking scheme with a *different* model. We then evaluate the dataset with O3 given the reference solution plus each alternative scheme. In both cases, performance degrades relative to using the original marking scheme across different metrics, suggesting that evaluator accuracy depends on close alignment with the marking scheme used by human although experts do not strictly follow them. Detailed results appear in §H.

## 4 DOWNSTREAM UTILITY: BEST-OF-N PROOF SELECTION

A good reward model should be able to reliably identify the best response in a given batch. To understand how this ability correlates with the calibration metrics in Section 3.2, we measure the best-of-$n$ (BoN) performance of various evaluators. In general, a higher BoN score means the evaluator more reliably selects high-quality candidates for subsequent training, guiding real downstream gains. The applications include selecting high-quality training data via rejection sampling (e.g., distillation) (Nakano et al., 2022; Touvron et al., 2023) and providing a reward signal for Reinforcement Learning (RL) (DeepSeek-AI et al., 2025; Yu et al., 2025).

In summary, this section answers a critical question: Does the superiority of our fine-grained evaluators, as measured by offline metrics, translate to an improvement in a downstream selection task?

**Setup.** To create a testbed, we use O3 to generate 16 candidate proofs for each of 29 selected problems from 2025, resulting in 464 unique proofs. All 464 of these candidates were then scored by three human experts using the pipeline described in §2. This dataset is distinct from PROOFBENCH and serves a different purpose. For each problem, 8 responses are graded by two experts with disagreements resolved through discussion. We test a selection of the fine-grained evaluators studied previously (§4.1) and several comparison-based selection strategies (§4.2). The performance of each selection method is measured using a **best-of-$n$ (BoN) curve**. Since an exhaustive evaluation over all $\binom{16}{n}$ subsets is computationally intensive, we estimate the BoN curve using Monte Carlo subsampling. For each $n$, we sample a large number of subsets of size $n$ without replacement. For each subset, the evaluator selects the single proof with the highest assigned score. The performance at $n$ is the average human score of these selected proofs, aggregated over all subsamples and all 29 problems. We also include two key baselines: an oracle baseline and a mean baseline. The HUMAN ORACLE represents the performance ceiling, calculated as the average score of the best possible selection among the sampled $n$ candidates. The MEAN BASELINE is calculated as the cumulative average score of the sampled candidates. All evaluators and selectors use O3 as the backbone model.

## 4.1 THE VALUE OF FINE-GRAINED SCORING

As shown in Figure 3a, our PROOFGRADER (an ensemble of five O3 runs with REF+MS) closely tracks the human-oracle curve, consistently selecting higher-quality proofs as $n$ increases. At $n$=16, it achieves an average score of 4.14. Comparing the four different fine-grained evaluators from §3: REF+MS ensemble (PROOFGRADER), REF+MS, REF, and NONE, we find the same ranking holds. Notably, NONE performs significantly worse than those using a marking scheme, confirming that the marking scheme is a critical component.

We further compare with binary evaluators: the O3-based models prompted to classify each proof as "Correct" or "Incorrect" rather than assigning a 0–7 score. In contrast to fine-grained evaluators, the binary evaluators perform much worse. This limitation comes from collapsing all "correct" (or "incorrect") proofs into a single category, losing the ability to rank solutions. When multiple correct candidates exist, it cannot distinguish an adequate proof (e.g., a 5/7) from an excellent one (7/7). We thus conclude that fine-grained scoring preserves relative ordering, making it essential for effective reward models in mathematical reasoning.

## 4.2 ROBUSTNESS AGAINST COMPARISON-BASED SELECTION STRATEGIES

Prior work (Liu et al., 2025b) explored relative comparison methods for improving best-of-$n$ (BoN), e.g., iteratively performing pairwise comparison between generations to select the best one. This is significantly more computationally expensive than using PROOFGRADER, which scores each candidate independently without pairwise evaluation. We compare PROOFGRADER against two comparison-based methods: Tournament (all-pairs) and Knockout (single-elimination).

**Tournament.** For candidates $s_1, \ldots, s_n$, the selector performs pairwise comparisons for every unordered pair $(i, j)$. Let $W_{ij} \in 0, 1$ indicate whether $s_i$ beats $s_j$. We form a win-rate matrix $W$ and compute $w_i = \sum_{j \neq i} W_{ij}$. The selected answer is $\arg\max_i w_i$ (tie-break by head-to-head or margin if applicable). This uses $\binom{n}{2}$ comparisons.

**Knockout.** Candidates are placed in a bracket; the selector compares pairs, and the winner advances until one remains. This requires $n - 1$ comparisons per bracket. Similar to other approaches, for each $n$, we average the scores over multiple random brackets.

Figure 3b shows that PROOFGRADER consistently dominates all selection variants across $n$, indicating that a strong, context-rich scoring function matters more than re-querying the same signal with pairwise procedures. Iterative strategies (Tournament, Knockout) still yield clear gains over mean baseline. Tournament may continue improving beyond $n$=16, but its quadratic cost $\mathcal{O}\left(\binom{n}{2}\right)$ is substantially higher than alternatives. Finally, as Tournament and Knockout can also be applied at test time to boost proof quality, the results suggest that proof generation benefits from increased test-time compute, echoing prior findings (Dekoninck et al., 2025).

## 5 RELATED WORK

**Benchmarks for Mathematical Reasoning.** GSM8K (Cobbe et al., 2021), MATH (Hendrycks et al., 2021), and AIME provide math reasoning benchmarks but mainly target short, closed-form answers. More recent competition-based benchmarks: Omni-MATH (Gao et al., 2024), Olympiad-Bench (He et al., 2024), HARP (Yue et al., 2024), and MathArena (Balunović et al., 2025) are more challenging yet still emphasize answer matching, leaving proof problems underrepresented. MathArena includes proofs but focuses only on latest competitions. Meanwhile, proof generation has advanced rapidly: Gemini (Luong & Lockhart, 2025), unreleased OpenAI systems, and model-agnostic verify-and-refine methods achieve IMO gold-level performance even for base models (Huang & Yang, 2025), but evaluations still mostly rely on manual grading (Petrov et al., 2025).

**Automated Evaluation for Generative Outputs.** LLM-as-a-judge methods use LLMs to score open-ended responses along dimensions such as correctness, enabling scalable evaluation while reducing human annotation costs (Zheng et al., 2023b; Li et al., 2024). Recent work has further examined the reliability of LLM judges on more challenging tasks (Tan et al., 2025; Frick et al., 2024). For mathematical proofs, however, automated evaluation remains relatively underexplored. Earlier work (Welleck et al., 2022; Frieder et al., 2023) often relies on human evaluation or lexical automatic metrics, which are difficult to scale and insufficient for verifying complex, competition-level proofs. Existing approaches are also often highly specialized, such as IneqMath (Sheng et al., 2025) for inequality proofs, or lack fine-grained evaluation; for example, the dataset of Dekoninck et al. (2025) provides only binary annotations. As a result, many studies still depend on manual evaluation for model-generated proofs (Petrov et al., 2025; Huang & Yang, 2025).

**Formal Proof Generation.** A complementary line of work uses interactive theorem provers, where LLMs generate proofs in formal languages (e.g., Lean, Isabelle) whose correctness can be automatically checked. Key benchmarks include miniF2F (Zheng et al., 2021), FIMO (Liu et al., 2023), PutnamBench (Tsoukalas et al., 2024), which formalize competition problems, and Lean-Workbook (Ying et al., 2024), a broad Lean corpus. While this approach is promising, our work focuses on informal proofs for two reasons. First, natural language remains the primary medium for mathematical communication in education and research. Second, the task of automatically translating informal proofs into a formal language, i.e., autoformalization, is itself an exceptionally challenging research problem (Gao et al., 2025; Liu et al., 2025a).

**Summary.** While prior work has established the importance of proof-based evaluation and explored the "LLM-as-a-judge" paradigm, a critical gap remains. No existing work has conducted a systematic, empirical study of the evaluator design space for mathematical proofs, nor provided a scalable methodology for creating the fine-grained, rubric-driven annotations necessary for such a study. The key factors that determine evaluator robustness remain largely uninvestigated.

## 6 CONCLUSION

We tackle the challenge of reliably evaluating natural-language mathematical proofs by introducing PROOFBENCH, the first comprehensive, fine-grained proof grading dataset spanning multiple contests and years. Using PROOFBENCH, we systematically explore the evaluator design space and show that strong backbones, problem-specific marking schemes, and ensembling are important, resulting in PROOFGRADER, our best-performing evaluator. Finally, we demonstrate practical utility by showing that using PROOFGRADER for best-of-$n$ selection closely tracks oracle performance.

**Limitations and Future Work.** While our work advances reliable evaluation of natural language math proofs, several limitations remain. First, our scope, i.e., olympiad-style proofs, does not yet cover research-level problems, or educational settings; extending to these regimes is an important next step. Second, evaluator quality can likely be improved through better prompting and workflow designs. Our dataset and analyses are intended as a foundation for such work, and we encourage future work to continue improving evaluators using PROOFBENCH as a testbed. Our evaluation focuses exclusively on mathematical correctness. Other crucial aspects such as readability, clarity, and elegance, are not currently assessed, and we leave the development of metrics for these qualities as future work. Third, our strongest evaluators currently rely on closed models. Although our design is model-agnostic, further improving fully open-source evaluators to match those based on closed models is an important direction. While our best-of-$n$ experiments use the evaluator as a selection signal, a first step toward training-time integration, we have not yet trained with it; incorporating it into proof generator training is a promising future direction.

ACKNOWLEDGEMENTS

We thank Robin Said Sharif and Bradley Louie for their invaluable contributions to the ProofBench dataset. We also thank Jingxuan He, Zhaoyu Li, and Tony Lian for valuable discussions during the early stages of this work, as well as Prasann Singhal and Aviral Kumar for their feedback on an earlier draft of the paper.

This work was partially supported by the Google BAIR Commons Program (2025–2026) and the Gemini Academic Program Award. Sky Computing Lab is supported by gifts from Accenture, Amazon, AMD, Anyscale, Broadcom, Google, IBM, Intel, Intesa Sanpaolo, Lambda, Lightspeed, Mibura, NVIDIA, Samsung SDS, and SAP.

ETHICS STATEMENT

Our work adheres to the ICLR Code of Ethics. The primary ethical considerations for this research involve the creation and use of our dataset, PROOFBENCH, and the potential for biases in our LLM-based evaluator. The dataset was constructed using problems from publicly accessible mathematics competitions. The expert annotations, which form the core of our benchmark, were provided by qualified individuals who were fairly compensated for their time and expertise.

REPRODUCIBILITY STATEMENT

We are committed to ensuring the reproducibility of our research. To this end, we will publicly release all code, data, and supplementary materials upon publication. The PROOFBENCH, including all problems, LLM-generated solutions, expert-annotated scores, and problem-specific marking schemes, will be made available. Our source code will include scripts to reproduce the experiments presented in this paper, including the implementation of our evaluator, PROOFGRADERand all prompts used.

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

## A    THE USE OF LARGE LANGUAGE MODELS (LLMs)

In preparing this submission, we used LLMs as a general-purpose writing assistant. The human authors were responsible for the entire research process, including the initial ideation, experimental design, implementation, and data analysis. During writing, authors first creating a complete draft of each section, which included all core ideas and technical details. The LLM was then used to revise this draft for clarity, conciseness, and grammatical correctness. Its role was strictly limited to improving the language and flow of the text. All scientific contributions and claims presented in this paper originate from the authors, who take full responsibility for the final content

## B    PROBLEMS INFORMATION

Table 6 provides short descriptions of the contests which we sourced problems from. The data information for the collected problems are available in Table 7. Problem 4 from EGMO 2023 was intentionally removed because it contains figure in problem text.

| Contest | Description | Source |
|---------|-------------|--------|
| APMO | The Asia Pacific Mathematical Olympiad is a regional proof-based contest for high school students across Asia–Pacific. It features five challenging problems solved over a fixed time window, proctored locally. | apmo-official.org |
| EGMO | The European Girls' Mathematical Olympiad is an international two-day proof contest for female students, modeled after the IMO. It promotes participation and excellence among young women in mathematics. | egmo.org |
| IMO | The International Mathematical Olympiad is the premier global high school math competition. Over two days, students tackle six proof problems that test creativity, rigor, and depth. | imo-official.org for problems and solutions from 2022 to 2025 and IMO 2025 problems. We use Evan Chen's Notes for IMO 2025 solutions. |
| PUTNAM | The William Lowell Putnam Mathematical Competition is a highly challenging proof exam for undergraduates in the U.S. and Canada. It consists of 12 problems emphasizing ingenuity and precise argumentation. | maa.org/putnam |
| TST | Team Selection Tests are national-level olympiad exams used by USA to select their IMO teams. Problems mirror international difficulty and assess readiness for global competition. | Evan Chen's Notes |
| USAMO | The United States of America Mathematical Olympiad is an invitational proof contest following AMC/AIME qualification. It identifies top U.S. high school problem solvers and feeds into further training and team selection. | Evan Chen's Notes |

Table 6: Brief descriptions of core contests.

## C    ANNOTATION PIPELINE DETAILS

### C.1    MARKING SCHEME GENERATION: EVALUATION PROCESS

We began by evaluating LLM-generated marking schemes with the objective of selecting an optimal configuration for reliable marking scheme generation. Two factors were considered: the backbone LLM and the prompting strategy (instructions).

| Contest | 2022 | 2023 | 2024 | 2025 | Total |
|---|---|---|---|---|---|
| APMO | 5 | 5 | 5 | 5 | 20 |
| EGMO | 6 | 5 | 6 | 6 | 23 |
| IMO | 6 | 6 | 6 | 6 | 24 |
| PUTNAM | 12 | 12 | 12 | – | 36 |
| TST | – | 6 | 6 | 6 | 18 |
| USAMO | 6 | 6 | 6 | 6 | 24 |
| **Total per year** | **35** | **40** | **41** | **29** | **145** |

Table 7: Problem counts by contest and year.

**Evaluation Protocol.** Throughout the evaluation process, two domain experts independently graded all marking schemes on a 0–3 scale using the rubric defined below. The experts compared two marking schemes at a time and rated both (the evaluation interface is shown in Figure 5).

**Scoring Guidelines:**

- **0**: The rubric is completely incorrect.

- **1**: The rubric seems reasonable, but the distinctions between scores and point allocations are not well-defined.

- **2**: The rubric is mostly correct, though details could be improved (e.g., inclusion of corner cases).

- **3**: The rubric is excellent—on par with those written by expert human graders.

We had the following experimental configuration:

**Models:** We compared two state-of-the-art models: OpenAI o3 and Gemini-2.5-Pro.

**Prompts:** Initial prompts were designed based on guidance from Evan Chen.[3] Chen's framework suggests organizing marking schemes into three sections: (1) a list of award points (indicating what progress merits credit), (2) a list of zero-credit points (for trivial observations), and (3) a list of deductions. The prompts were also designed to enforce consistent output formatting.

We compared zero-shot and few-shot prompting strategies. For few-shot prompting, examples were selected from Evan Chen's rubrics for the United States Emergent Mathematicians Olympiad (USEMO),[4] as marking schemes from other competitions are not publicly available.

**Stage 1 (Model Selection):** We evaluated 18 problems, yielding 36 marking schemes (one per model). Zero-shot prompting was used for all generations. The two annotators differed by more than 1 point in only three cases. Overall, they preferred Gemini-2.5-Pro. Based on annotator feedback, we revised the prompts accordingly.

**Stage 2 (Prompt Selection):** We evaluated 36 problems, comparing zero-shot and few-shot prompting strategies. Both experts rated zero-shot prompting slightly better. Specifically, Expert 1 rated zero-shot as better in 13/36 cases, worse in 10/36 cases, and equal in 13/36 cases; Expert 2 rated zero-shot as better in 17/36 cases, worse in 8/36 cases, and equal in 11/36 cases. Feedback from Stage 2 was used to further refine the prompt design.

**Average Rubric Quality:** Of the 36 rubrics generated in Stage 2, 35 were rated 2 or 3 by both annotators, indicating high overall quality. Note that we did not generate or evaluate rubrics for all available problems.

**Final Configuration.** The final configuration was selected based on both quantitative annotator scores and qualitative feedback. While no single setup produced perfect rubrics across all problems, we chose the configuration with the best overall performance: **Gemini-2.5-Pro with zero-shot prompting**.

---

[3] https://web.evanchen.cc/static/usemo/captain-guidance-usemo.pdf
[4] https://web.evanchen.cc/usemo.html

**Annotator Feedback.** We include representative feedback from the annotators below to provide insight into the rubric quality and areas for improvement:

---

**Expert Annotator Comments**

**General observations:**

- Several problems included deductions for minor calculation errors, despite the prompt explicitly instructing against this practice. Due to the frequency of this issue, we did not comment on every occurrence.

- Problems repeated from previous batches retained preloaded feedback from earlier evaluations. Clearing this feedback between batches would prevent confusion about which problems have been completed versus those requiring re-evaluation.

- Many problems awarded partial credit merely for stating a lemma, definition, or formula without requiring proof or demonstration of its significance. This was the most significant issue to address in future iterations.

- Partial credit allocation typically followed the official solution path rather than recognizing alternative approaches that yield weaker (but still meaningful) results. For example, in USEMO-2024-P1, the AI-generated rubrics allocated partial credit across steps in the official solution, whereas the official marking scheme (by Evan Chen) awarded partial credit based on the strength of justified bounds. This discrepancy arises because the model cannot anticipate alternative solution strategies without explicit examples. A prompt modification could encourage such flexibility—for instance: "prove $X$ bound [max 4] (step 1 [additive 1], step 2 [additive 3]) (proving weaker bound $Y$ earns 2 points partial)."

- The "additive up to 7" scoring feature in the generated schemes does not align well with available contest rubrics. Whether this is acceptable depends on the overall objectives of the evaluation system.

---

An example of marking scheme together with the problem and reference solutions is shown in Figure 4.

## C.2 EVALUATION OF MODEL-GENERATED PROOFS

**Pilot Calibration.** Using the frozen rubric generator, we produced problem-specific marking schemes and calibrated the scoring protocol. Two experts independently annotated 36 problems with 3 responses each (108 total solutions). The experts then discussed disagreements to reach consensus and refine their grading protocol accordingly. They will share their experience with other experts.

**Annotation Process.** Throughout this study, two datasets were evaluated by a team of five experts: ProofBench and the dataset used in the best-of-$n$ experiments. For each solution, experts were provided with the problem statement, reference solutions, the corresponding marking scheme, and an optional AI-generated judgment to inform their final grade on a 0–7 scale. Screenshots of the annotation interface are shown in Figures 6 and 7.

**Double Grading.** To ensure annotation quality, more than 40% of solutions in both datasets were independently graded by two experts. Specifically:

- **ProofBench**: All problems from 2025 competitions, as well as all EGMO (European Girls' Mathematical Olympiad) and APMO (Asian Pacific Mathematics Olympiad) problems, were double-graded.

- **Best-of-$n$ datasets**: For each problem, 8 out of 16 responses were double-graded.

The annotation process was conducted in dedicated meeting rooms with experts working collaboratively. The entire process was supervised by a PhD student and a professor (both co-authors of this work).

**Disagreement Resolution.** Disagreements were flagged when expert ratings differed by more than 1 point. In such cases, the two experts discussed their rationales and justifications. On the 0–7 scale,

**Problem.** Let $n > k \geq 1$ be integers. Let $P(x) \in \mathbb{R}[x]$ be a polynomial of degree $n$ with no repeated roots and $P(0) \neq 0$. Suppose that for any real numbers $a_0, \ldots, a_k$ such that $a_k x^k + \cdots + a_1 x + a_0$ divides $P(x)$, the product $a_0 a_1 \ldots a_k$ is zero. *Prove* that $P(x)$ has a nonreal root.

**Reference Solution.** By considering any $k+1$ roots of $P$, WLOG assume $n = k+1$. Suppose $P(x) = (x + r_1) \ldots (x + r_n)$ has $P(0) \neq 0$. Then each polynomial $P_i(x) = P(x)/(x + r_i)$ of degree $n-1$ has $\geq 1$ zero coefficient.
The leading and constant coefficients of each $P_i$ are nonzero, leaving $n-2$ other coefficients. By pigeonhole, $P_1$ and $P_2$ share a zero coefficient position, say $x^k$ for some $1 \leq k < n-1$.

*Claim.* If $P_1$ and $P_2$ both have $x^k$ coefficient zero, then $Q(x) = (x + r_3) \ldots (x + r_n)$ has consecutive zero coefficients $b_k = b_{k-1} = 0$.

*Proof.* By Vieta, let $Q(x) = x^{n-2} + b_{n-3} x^{n-3} + \cdots + b_0$. The $x^k$ coefficient of $P_1, P_2$ being zero means $r_1 b_k + b_{k-1} = r_2 b_k + b_{k-1} = 0$,
hence $b_k = b_{k-1} = 0$ (using $r_i$ nonzero, distinct). $\square$

*Lemma.* If $F(x) \in \mathbb{R}[x]$ has two consecutive zero coefficients, it cannot have all distinct real roots.

*Proof 1 (Rolle).* Say $x^t, x^{t+1}$ coefficients are zero. If all roots are real and distinct, Rolle's theorem implies every derivative has this property.
But $F^{(t)}(x)$ has a double root at 0, contradiction. $\square$

*Proof 2 (Descartes).* Real roots are bounded by sign changes in $F(x)$ plus sign changes in $F(-x)$. For consecutive nonzero coefficients $\star x^i, \star x^j$ $(i > j)$: if $i - j = 1$, the sign change counts once; if $i - j \geq 2$, it may count twice but there's $\geq 1$ zero between them.

With $b$ nonzero coefficients and $z$ runs of zeros, real roots $\leq b - 1 + z \leq \deg F$. Two consecutive zeros make this strict. $\square$

**Marking Scheme (max 7 pts).** *Checkpoints (additive):*
**(1)** [1pt] Problem reduction to $n = k+1$ and setup. Alt: complete proof for $k = 1$.
**(2)** [2pts] Pigeonhole: $n$ polynomials, $n-2$ internal coefficient positions; two share a zero position.
**(3)** [2pts] Deduce consecutive zeros: from $[x^m]P_1 = [x^m]P_2 = 0$ and $P_i = (x + r_i)Q$, show $b_m = b_{m-1} = 0$.
**(4)** [2pts] Prove lemma (2pts for complete Rolle or Descartes proof; 1pt for partial).
*Deductions:* Cap 6/7 if no reduction justification; cap 5/7 if lemma flawed; cap 3/7 if stops after PHP; $-1$pt for minor gaps (e.g., not using $r_1 \neq r_2$).
*Zero credit:* Unjustified WLOG; merely stating theorems; specific examples only; noting $P(0) \neq 0 \Rightarrow$ nonzero roots.

Figure 4: Example problem (USAMO 2025 P2) with its reference solution and marking scheme.

scores naturally cluster into four bands: incorrect (0), partial progress (1–3), nearly complete (4–6), and fully correct (7). During discussion, annotators must agree on (1) which band a solution belongs to, and (2) the exact score for fully correct (7) and fully incorrect (0) solutions. For partial-credit cases (1–6), minor differences may initially arise due to inherent ambiguities in natural-language proofs and subjective judgment even with marking scheme guidance, but these are resolved through further discussion. All scores in the final dataset reflect this consensus-based adjudication.

**Annotator Assignments.** Annotation responsibilities were distributed as follows:

- **ProofBench**: Solutions were annotated by Experts A, B, and C.
- **Best-of-$n$ datasets**: Solutions were primarily annotated by Experts D and E, with support from Expert B.
- **Marking schemes**: Rubrics were evaluated by Experts A and B.

## D   PROOFBENCH ABLATION BY COMPETITION AND YEAR

We analyze potential data contamination by comparing model performance on problems from competitions occurring before versus after their respective knowledge cutoff dates. As shown in Table 9, all three models exhibit positive bias toward pre-cutoff problems: Gemini-2.5-pro ($\Delta = +1.01$, significant), OpenAI-o3 ($\Delta = +0.44$, moderate), and DeepSeek-R1-0528 ($\Delta = +0.40$, moderate).

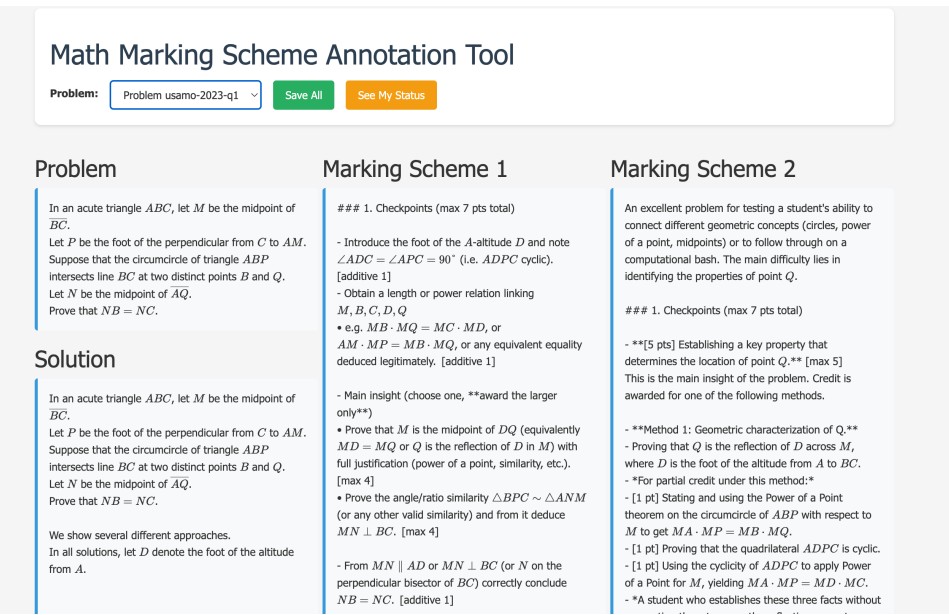

Figure 5: **View of Marking Evaluation Platform Setup**

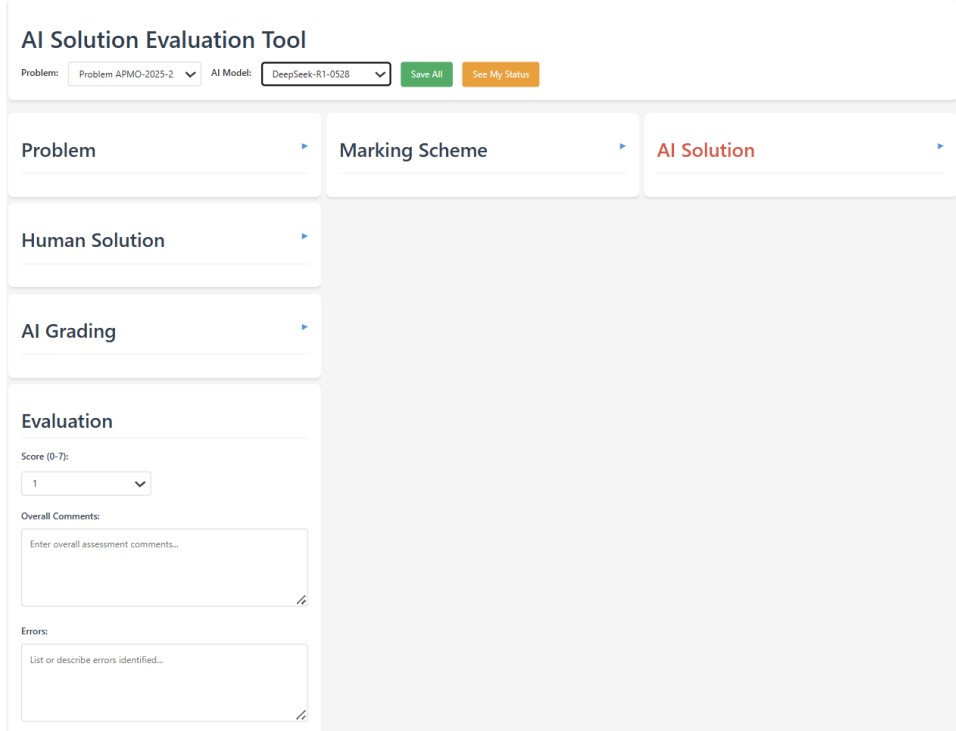

Figure 6: **View of Evaluation Platform Setup**

Gemini-2.5-pro demonstrates the strongest contamination signal (14.4% performance improvement on pre-cutoff data), which is attributable to its late cutoff date (January 1, 2025) resulting in only 20% of our dataset being clean evaluation data. Table 8 reveals heterogeneous contamination patterns across competitions: while IMO shows consistent positive contamination across all models, other competitions like EGMO and APMO exhibit mixed or reverse effects, suggesting interactions between contamination and problem difficulty variations.

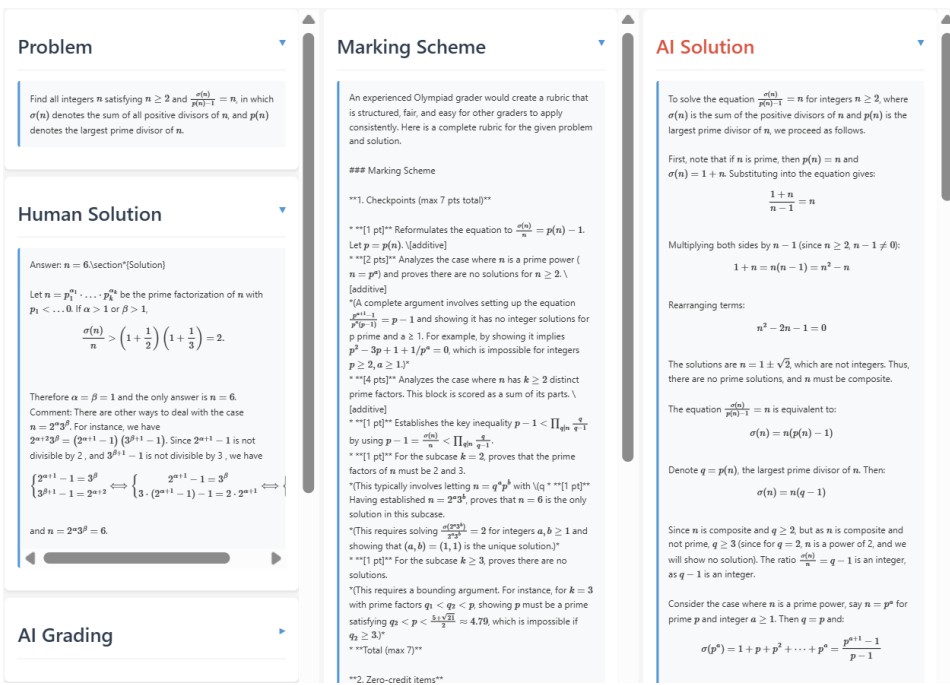

Figure 7: **View of Evaluation Platform Setup**

To mitigate contamination effects, we focus our primary analysis on the 29 problems from 2025 competitions (APMO, EGMO, IMO, USAMO, TST 2025), which provide clean evaluation data for all three models. While this reduces our sample size to 20% of the full dataset, it ensures fair cross-model comparison without training data leakage confounds. We report full dataset results for completeness but mark pre-cutoff data with contamination warnings. These findings highlight the need for continuously updated benchmarks using very recent competitions (within 3–6 months) and increased transparency from model developers regarding training data composition and cutoff dates.

## E   ADDITIONAL DETAILS FOR EVALUATION METRICS

**Within-problem ranking agreement.**   For problem $p$, consider all pairs $(i, j)$ with $i < j$. Define $\Delta_{ij}^{\exp} = y_{pi} - y_{pj}$ and $\Delta_{ij}^{\text{eval}} = \hat{y}_{pi} - \hat{y}_{pj}$. Let $C$ and $D$ be the numbers of concordant and discordant pairs, respectively, and let

$$T_{\exp} = \#\{(i, j) : \Delta_{ij}^{\exp} = 0\}, \qquad T_{\text{eval}} = \#\{(i, j) : \Delta_{ij}^{\text{eval}} = 0\}.$$

Kendall's $\tau_b$ for problem $p$ is

$$\tau_b(p) = \frac{C - D}{\sqrt{(C + D + T_{\exp})(C + D + T_{\text{eval}})}},$$

and is undefined if the denominator is zero (we omit such $p$ from aggregation).

**Macro averaging.**   We macro-average per-problem metrics across problems. Writing $\mathcal{P}$ for the set of problems with defined $\tau_b$ and $P' = |\mathcal{P}|$:

$$\overline{\text{MAE}} = \frac{1}{P} \sum_{p=1}^{P} \text{MAE}_p, \quad \overline{\text{RMSE}} = \frac{1}{P} \sum_{p=1}^{P} \text{RMSE}_p, \quad \overline{\text{Bias}} = \frac{1}{P} \sum_{p=1}^{P} \text{Bias}_p,$$

$$\overline{\text{WTA}}(\leq 1) = \frac{1}{P} \sum_{p=1}^{P} \text{WTA}_p(\leq 1), \overline{\tau_b} = \frac{1}{P'} \sum_{p \in \mathcal{P}} \tau_b(p).$$

(As noted in the main text, all problems have the same response count $n$.)

| Competition | Year | | | | Summary | | | | |
|---|---|---|---|---|---|---|---|---|---|
| | 2022 | 2023 | 2024 | 2025 | Contam. | Clean | Mean Before | Mean After | Δ |
| **OpenAI-o3** (cutoff: June 1, 2024) | | | | | | | | | |
| APMO | × 3.8 n=5 | × 4.0 n=5 | × 3.2 n=5 | ✓ 4.6 n=5 | 15 | 5 | 3.67 | 4.60 | -0.93 |
| EGMO | × 4.2 n=5 | × 3.4 n=6 | × 3.5 n=6 | ✓ 5.2 n=6 | 17 | 6 | 3.71 | 5.17 | -1.46 |
| IMO | × 2.8 n=6 | × 2.7 n=6 | ✓ 2.0 n=6 | ✓ 1.0 n=6 | 12 | 12 | 2.75 | 1.50 | +1.25 |
| PUTNAM | × 3.3 n=12 | × 3.4 n=12 | ✓ 3.0 n=12 | – no data | 24 | 12 | 3.33 | 3.00 | +0.33 |
| TST | – no data | × 1.7 n=6 | ✓ 1.3 n=6 | ✓ 1.2 n=6 | 6 | 12 | 1.67 | 1.25 | +0.42 |
| USAMO | × 2.3 n=6 | × 1.8 n=6 | × 2.0 n=6 | ✓ 2.3 n=6 | 18 | 6 | 2.06 | 2.33 | -0.28 |
| **Total** | | | | | **92** | **53** | **3.02** | **2.59** | **+0.44** |
| **Gemini-2.5-pro** (cutoff: January 1, 2025) | | | | | | | | | |
| APMO | × 3.2 n=5 | × 2.8 n=5 | × 3.2 n=5 | ✓ 3.6 n=5 | 15 | 5 | 3.07 | 3.60 | -0.53 |
| EGMO | × 3.6 n=5 | × 2.8 n=6 | × 3.0 n=6 | ✓ 1.0 n=6 | 17 | 6 | 3.12 | 1.00 | +2.12 |
| IMO | × 2.5 n=6 | × 2.2 n=6 | × 2.0 n=6 | ✓ 1.5 n=6 | 18 | 6 | 2.22 | 1.50 | +0.72 |
| PUTNAM | × 3.1 n=12 | × 3.3 n=12 | × 3.2 n=12 | – no data | 36 | 0 | 3.19 | – | – |
| TST | – no data | × 2.2 n=6 | × 2.0 n=6 | ✓ 1.2 n=6 | 12 | 6 | 2.08 | 1.17 | +0.92 |
| USAMO | × 2.5 n=6 | × 2.0 n=6 | × 1.8 n=6 | ✓ 1.7 n=6 | 18 | 6 | 2.11 | 1.67 | +0.44 |
| **Total** | | | | | **116** | **29** | **2.73** | **1.72** | **+1.01** |
| **DeepSeek-R1-0528** (cutoff: July 1, 2024) | | | | | | | | | |
| APMO | × 2.6 n=5 | × 2.2 n=5 | × 2.2 n=5 | ✓ 1.4 n=5 | 15 | 5 | 2.33 | 1.40 | +0.93 |
| EGMO | × 3.0 n=5 | × 2.2 n=6 | × 2.2 n=6 | ✓ 1.5 n=6 | 17 | 6 | 2.41 | 1.50 | +0.91 |
| IMO | × 1.5 n=6 | × 1.2 n=6 | ✓ 0.8 n=6 | ✓ 0.3 n=6 | 12 | 12 | 1.33 | 0.58 | +0.75 |
| PUTNAM | × 2.8 n=12 | × 2.5 n=12 | ✓ 3.4 n=12 | – no data | 24 | 12 | 2.63 | 3.42 | -0.79 |
| TST | – no data | × 0.7 n=6 | × 0.5 n=6 | ✓ 1.0 n=6 | 12 | 6 | 0.58 | 1.00 | -0.42 |
| USAMO | × 2.0 n=6 | × 1.5 n=6 | × 1.3 n=6 | ✓ 0.5 n=6 | 18 | 6 | 1.61 | 0.50 | +1.11 |
| **Total** | | | | | **98** | **47** | **1.95** | **1.55** | **+0.40** |

Table 8: Contamination status by competition, year, and model. Each cell shows mean expert rating (1-7 scale) and problem count. × = potentially contaminated (before cutoff), ✓ = clean (after cutoff), – = no data. Summary columns show contamination effect (Δ = before – after).

## F EVALUATION ON WEAKER, OPEN-SOURCE MODELS

We collected expert gradings on solutions from two weaker models (Qwen3-4B and DeepSeek-R1-0528-Qwen3-8B) on a selected set of 36 problems from 2024 and 2025 competitions. Both models are among the best math reasoning models of the same scale. Here we provide the results on the generators (Table 11) and selected evaluators (Table 12).

| Model | Cutoff Date | Contaminated (#) | Clean (#) | Mean Before | Mean After | Δ |
|---|---|---|---|---|---|---|
| OpenAI-o3 | June 1, 2024 | 92 (63%) | 53 (37%) | 3.02 | 2.59 | +0.44 |
| Gemini-2.5-pro | Jan 1, 2025 | 116 (80%) | 29 (20%) | 2.73 | 1.72 | +1.01 |
| DeepSeek-R1-0528 | July 1, 2024 | 98 (68%) | 47 (32%) | 1.95 | 1.55 | +0.40 |

Δ = Mean difference (contaminated – clean); positive values indicate contamination.
Expert ratings on 1–7 scale; dataset contains 145 problems × 3 solutions = 435 total.

Table 9: Contamination analysis summary across three models. All three show positive bias toward pre-cutoff data, with Gemini-2.5-pro exhibiting the strongest effect.

| Contest | OpenAI-o3 | Gemini-2.5-pro | DeepSeek-R1-0528 |
|---|---|---|---|
| APMO | -0.93 | -0.53 | +0.93 |
| EGMO | -1.46 | +2.12 | +0.91 |
| IMO | +1.25 | +0.72 | +0.75 |
| PUTNAM | +0.33 | – | -0.79 |
| TST | +0.42 | +0.92 | -0.42 |
| USAMO | -0.28 | +0.44 | +1.11 |
| **Overall** | **+0.44** | **+1.01** | **+0.40** |

Δ values: strong contamination (>1.0), moderate (0.3–1.0), no contamination or reverse effect (<0.3)

Table 10: Contest-specific contamination effects (Δ) across models. Red indicates stronger performance on pre-cutoff data (contamination), green indicates better performance on post-cutoff data.

PROOFGRADER's error rates are actually significantly lower here than those reported in the main paper. This is because weaker models mostly produce incorrect (0-point) proofs, making grading much easier. Since grading weaker models is generally easier, we chose to use the strongest available model as generators in PROOFBENCH. This allows the benchmark to better challenge evaluators by requiring them to distinguish between partially correct and nearly-complete solutions, and to detect subtle errors in reasoning steps.

| Generator | Average score (out of 7) | 0-point solutions |
|---|---|---|
| Qwen3-4B | 0.92 | 70% |
| DeepSeek-R1-0528-Qwen3-8B | 0.78 | 72% |

Table 11: Weaker generator performance on selected problems.

# G   WEAKER MODELS AS EVALUATORS

We have conducted additional experiments with leading open-source models at different capability levels: Qwen3-235B-A22B (Thinking) and Llama-3.1-70B-Instruct.

Results are shown in Table 13. Unfortunately, these models demonstrate substantially weaker performance as evaluators compared to frontier proprietary models. Given that these represent some of the strongest available open-source models in their respective families, evaluating smaller variants (e.g., 7B or 14B models) would likely yield even weaker results. These findings suggest that current open-source models significantly lag behind proprietary models in fine-grained mathematical proof evaluation.

| Evaluator | MAE | RMSE | WTA($\leq$1) |
|---|---|---|---|
| O3 ensemble (Ref+MS) – ProofGrader | 0.431 | 0.521 | 93.1% |
| o3 (Ref+MS) | 0.486 | 0.559 | 95.8% |
| o3 (None) | 1.764 | 1.892 | 59.7% |
| DeepSeek-R1(Ref+MS) | 0.750 | 0.929 | 81.9% |
| DeepSeek-R1(None) | 3.653 | 3.823 | 11.1% |
| Gpt-4o (Ref+MS) | 2.458 | 2.669 | 26.4% |
| Gpt-4o (None) | 3.931 | 4.056 | 5.6% |

Table 12: Evaluator performance metrics on weaker generators.

| Model | Context | MAE | RMSE | Bias | WTA($\leq$1) |
|---|---|---|---|---|---|
| Llama-3.1-70 B-Instruct | Ref+MS | 3.212 | 3.679 | 3.156 | 31.0% |
| | MS | 3.189 | 3.385 | 3.150 | 27.9% |
| | Ref | 3.194 | 3.661 | 3.096 | 31.2% |
| | None | 3.945 | 4.404 | 3.898 | 26.1% |
| Qwen3-235B -A22B | Ref+MS | 1.531 | 2.007 | 0.724 | 65.3% |
| | MS | 1.351 | 1.727 | 0.041 | 66.8% |
| | Ref | 1.860 | 2.068 | 1.425 | 45.3% |
| | None | 2.631 | 3.070 | 2.359 | 40.6% |

Table 13: Model performance comparison across different contexts for two additional models: Llama-3.1-70B-Instruct and Qwen3-235B-A22B.

## H    IN-DEPTH ANALYSES

**What are the common failure modes of PROOFGRADER?**    After examining PROOFGRADER's outputs, we find that cases of over-crediting and under-crediting by more than one point are relatively balanced (10.8% vs. 12.2%). Across generators, PROOFGRADER makes the most errors on o3-generated solutions–28.9% of the solutions, compared to 23.3% for GEMINI and 16.9% for GEMINI. Since PROOFGRADER itself is based on O3, this suggests a degree of within-generator bias. By problem type, PROOFGRADER fails more frequently on Algebra and Geometry problems (each accounting for $\sim$25% of failure cases) than on other domains. For qualitative analysis, we manually examined 50 solutions for which two or more ensemble runs of PROOFGRADER deviated from expert scores by at least two points. This reveals several common failure modes:

**Overcrediting:**

- **Appearance-of-completeness.** PROOFGRADER sometimes assigns high scores to solutions whose structure closely mirrors the rubric (sections, lemmas, final condition) even when the central mathematical claim is false, so superficial alignment masks a fatally incorrect core argument.

  *Case Study:* o3 - APMO-2025-2. The solution performs several correct manipulations but incorrectly rewrites a key condition so that a range of $(a, b)$ values is claimed to always satisfy the constraint, even though that is false.

- **Fatal gaps treated as minor omissions.** When a proof relies on a false universal claim or a broken reduction, PROOFGRADER often interprets this as "missing justification" rather than a fatal error, granting partial credit even though the entire method collapses.

  *Case Study:* R1 – APMO 2024/3. R1 claims "$f(s, t)$ is always nonnegative", but direct counterexamples show $f(s, t) < 0$. This invalidates the whole reduction.

- **Overtrust in sophisticated but incorrect frameworks.** For solutions that use polished frameworks (coordinates, height functions, custom transformations), PROOFGRADER tends to reward the global structure and technical vocabulary without verifying key validity conditions, leading to substantial overcredit for incomplete or invalid arguments.

*Case Study:* Gemini – APMO-2023-5. Gemini constructs an elaborate height-function argument but never verifies that paths do not cross or that the constructed ordering is actually consistent, which is required for the proof.

**Undercrediting:**

- **Penalizing early missteps despite a correct final proof.** LLMs often explore an incorrect approach before restarting with a valid proof; humans grade the final argument, but PROOF-GRADER aggregates all attempts and heavily penalizes early errors, leading to near-zero scores even when the final proof is fully correct.

  *Case Study:* R1 – APMO 2023/2. R1 initially writes incorrect divisor equations and claims "larger $k$ gives no solution." Then it restarts and produces a valid classification.

- **Double-penalizing a single flaw.** A single conceptual error can both prevent multiple checkpoints from firing and trigger a global "major error" deduction, so partially correct solutions with substantial progress can be driven all the way to 0/7.

  *Case Study:* R1 – PUTNAM 2022 B4. The solution analyzes several structural cases correctly but ends with an incorrect final expression that contradicts its earlier reasoning.

- **Overstrict demand for micro-justifications.** Omitted trivial steps or minor slips (e.g., a mislabeled point) can cause PROOFGRADER to mark long downstream chains as unproven, heavily undergrading solutions whose main ideas and overall argument would receive high partial credit from human graders.

  *Case Study:* o3 – USAMO 2023/1. The solution has a valid coordinate approach but mislabels a point, causing PROOFGRADER to treat all downstream angle and collinearity relations as missing justification.

**Geometry Instability.** Geometry problems are particularly brittle: long dependency chains (similarity $\rightarrow$ ratios $\rightarrow$ concurrency/cyclicity) mean a single incorrect or missing step can either be wrongly ignored (leading to overcredit, e.g., treating a non–circle-preserving transformation as valid) or over-amplified (leading to undercredit when a routine angle equality is left implicit), producing highly unstable scores on geometric proofs (e.g., o3 – APMO 2024/1 vs. o3 – EGMO 2025/4).

**Why are reference solutions and marking schemes helpful?** Comparing evaluators based on o3 with NONE to those given a REF+MS, we find that in over $60\%$ of cases, the evaluator without context *overestimates* the generation's correctness, i.e., it incorrectly assigns a *higher* score to the generation. Crucially, this overestimation is not uniform: we observe a strong correlation ($r = 0.699$, $p < 0.001$) between proof quality and evaluation gap. For *low-quality* proofs (scored 0–2), the no-context evaluator over-scores by an average of 1.7 points. In contrast, for *high-quality* proofs (scored 5–7), the evaluator with context information assigns *higher* scores by 0.8 points on average. This asymmetric pattern supports our hypothesis that, without reference solutions or marking schemes, the evaluator struggles to gauge proof *progress*.

This effect is exacerbated when the evaluator cannot fully solve the problem, a case that constitutes the majority of our dataset (as shown in Figure 2). When we examine problems that o3 cannot solve well (scoring $\leq 2$ when acting as a prover), we find that the evaluator without context over-scores by 1.4 points on average. For problems that o3 can solve (scoring $\geq 5$), the evaluator with context scores higher by 0.9 points. This demonstrates that the evaluator's difficulty in solving a problem directly impairs its ability to assess partial progress on that problem without reference.

**Case study.** On Putnam-2022-B5 (generator: o3), the evaluator with context identified a critical logical flaw ("log-concavity inequality fails when $|a_i| > 1$") and assigned 1 point, while the no-context evaluator deemed the proof "essentially correct" and awarded 7 points. Manual inspection of the 20 cases with largest disagreements ($|\Delta| \geq 4$) reveals that 80% involve proofs with sophisticated presentation but fundamental logical errors, supporting the view that reference solutions help evaluators distinguish between fluent exposition and mathematical correctness.

**Sensitivity to Marking Schemes.** Table 14 reports results for o3 evaluators under REF+MS, comparing three marking-scheme variants: (i) the original schemes used by human experts, (ii)

| Model | Marking Scheme | RMSE↓ | MAE↓ | WTA$_{\leq 1}$ (%)↑ | Kendall-$\tau$ ↑ | Bias≈ |
|---|---|---|---|---|---|---|
| O3 | **Original** | **1.273** | **0.964** | **76.5** | 0.502 | **−0.008** |
| | New marking scheme (GEMINI) | 1.467 | 1.156 | 70.0 | **0.515** | 0.219 |
| | New marking scheme (O3) | 1.525 | 1.196 | 70.4 | 0.460 | 0.020 |

Table 14: **Marking Scheme Sensitivity.** We evaluate the single-pass O3 evaluator under REF+MSwith three marking-scheme sets—original (human), regenerated (same model/prompt), and O3-generated (same prompt). The original human schemes perform best.

schemes regenerated by the same model with the same prompt, and (iii) schemes generated by O3 itself with the same prompt. The original human-provided schemes yield the strongest performance.

# I GENERATOR PROMPT

> **Default**
>
> ```
> Your task is to write a proof solution to the following
> problem. Your proof will be graded by judges for correctness
> and completeness. When you write your proof, follow these
> guidelines:
> - You are creating a proof, not a proof outline. Each step
> should be carefully explained and documented. If not properly
> explained, the judge will assume that you cannot explain it,
> and therefore decrease your grade.
> - You can use general theorems and lemmas, but only if they
> are well-known. As a rule of thumb: if the result has a name
> and is famous enough to have a Wikipedia page or something
> similar to describe it, it is allowed. Any result from papers
> that would not be taught in high school or low-level bachelor
> courses in mathematics should not be used. Any use of such
> results will immediately give you a zero grade.
> - Do not skip computation steps in your proof. Clearly
> explain what transformations were done and why they are
> allowed in each step of a calculation.
> - Your proof should be self-contained.
> - If you are not sure about a specific step, or do not
> know how to prove an intermediate result, clearly state
> this. It is much preferable to indicate your uncertainty
> rather than making incorrect statements or claims.
>
> FORMATTING GUIDELINES:
> - You should write Markdown with LaTeX math. Do NOT use
> code fences (no ```).
> - You should use correct LaTeX notation to write equations
> and mathematical symbols. You should encompass these
> equations in correct delimiters ("\\(" and "\\)" for
> inline math, "\\[" and "\\]" for block math) to enhance
> the clarity of your proof. **Do not use any unicode
> characters.**
> - For multi-line derivations, wrap an aligned block
> INSIDE display math.
> - Do not use other LaTeX environments or packages.
>
> PROBLEM: {problem}
> ```

## J   Marking Scheme Generation Prompt

---

**Marking Scheme (without examples)**

You are an experienced Olympiad grader.
*Treat the official solution as fully correct and
authoritative; do not claim it contains errors or gaps.*
Your task is to write a complete, grader-friendly rubric for
the problem and official solution below. The rubric must map
a student's proof to an integer score from **0 to 7**.

---

### INSTRUCTIONS

Produce the marking scheme in **exactly** the following three
sections.
1. **Checkpoints (max 7 pts total)**

  * Break the solution into logically independent checkpoints
  with **integer** point values.

    * Allocate **>= 4 pts** to the main idea/critical steps;
    **<= 3 pts** to routine work.
  * If two items are mutually exclusive (solve the *same*
  logical gap), **nest** them and write **\award the larger
  only"**.
  * **Parallel solution paths (non-additive):**
    * If the official solution (or a student submission)
    admits more than one legitimate approach, write
    **parallel checkpoint chains** labeled \Chain A
    / Chain B / ... (idea: ...)".
    * **Start this section with a bold rule:** **Score
    exactly one chain | take the **maximum** subtotal
    among chains; do **not** add points across chains.**
    * Within a chain, checkpoints may be \[additive]. For
    steps **shared across chains** (e.g., a lemma usable by
    multiple approaches), place them under a \Shared
    prerequisites" group with **\[max k]**, and state
    **\count at most once regardless of chain."**
  * For every bullet (or group), append either
  **\[additive]** or **\[max k]** to make the scoring
  rule unambiguous.
  * Finish with a one-line **\Total (max 7)"** check that is
  consistent with the non-additivity rule.
  * Never demand cosmetic labels or specific variable names
  unless essential to the logic.

2. **Zero-credit items**

  * List common arguments or observations that **earn 0
  points** (e.g., conjectures without proof|especially in
  geometry, routine restatements of the problem, or
  dead-ends).

3. **Deductions**

  * Bullet each typical mistake with a **flat** penalty
  (**{1**, **{2**, or **\cap at x/7"**).
  * Apply **at most the single largest** deduction; never
  reduce a score below 0.
  * Target logic gaps, invalid claims, circular reasoning,
  or contradictions. Cosmetic slips (notation, arithmetic,
  wording) do **not** trigger deductions unless they break

```
   validity.
   * If a student gives **multiple distinct proofs**, **grade
   the best one only** (no stacking). If proofs contain
   **contradictory claims**, apply an appropriate deduction
   (e.g., **cap at 5/7**).

### IMPORTANT REQUIREMENTS

* Use **concise bullets** | no prose exposition of the
official solution.
* **Arithmetic sanity checks:** per-chain checkpoint sums
and \[max k]
caps must make it impossible to exceed **7** overall; at
least one chain must allow a perfect **7/7**.
* Do **not** introduce, \fix," or critique the official
solution.
* Avoid over-fragmenting: do not split routine algebra
into 3+ separate 1-pt bullets.
* Keep notation consistent with the official solution;
define any new symbols you introduce.

─────────────────────────────────────────────────────

### PROBLEM

{problem}

### OFFICIAL SOLUTION

{solution}
```

### Marking Scheme (with examples)

```
You are an experienced Olympiad grader.

Your task is to write a complete, grader-friendly rubric for the
problem and official solution given below.
*Treat the official solution as fully correct and authoritative;
do not claim it contains errors or gaps.*

The rubric must map a student's proof to an integer score from 0 to 7.

─────────────────────────────────────────────────────

### INSTRUCTIONS

Produce the marking scheme in **exactly** the following three sections.

1. Checkpoints (max 7 pts total)
– Break the solution into logically independent checkpoints.
– Assign each checkpoint an integer value.
    – Allocate >= 4 pts total to the main part of the proof;
    <= 3 pts total to routine work.
– If two items are mutually exclusive (solve the same gap), nest them
and write**"award the larger only"**.
– For every bullet, append either [additive] or [max k] to make the
scoring rule unambiguous.
– Finish the section with a one-line \Total (max 7)" check.
– If the official solution mentions more than one legitimate path,
give either (a) parallel checkpoint chains or (b) a catch-all
checkpoint worth up to 7 pts for a completely correct alternative
proof that uses the same underlying idea.
– Never demand cosmetic labels or a specific variable naming unless
```

```
they are essential to the logic.

2. Zero-credit items
List common arguments or observations that look related but
**earn 0 points**. Include conjectures stated with no proof
(esp. geometry), routine restatements, or dead-end explorations.

3. Deductions
- Bullet each typical mistake with a flat penalty:
-1, -2, or \cap at x/7".
- Apply at most the single largest deduction; never reduce a score
below 0.
- Penalties should target logic gaps, invalid claims, or circular
reasoning. Cosmetic slips (notation, arithmetic, wording) never trigger
deductions unless they break validity.

### IMPORTANT REQUIREMENTS
- Use concise bullet points; no prose exposition of the official
solution.
- Arithmetic sanity checks: checkpoint points + exclusivity rules must
sum to 7. The rubric should be self-evident to a grader with no
calculator.
- Do not introduce, \fix," or critique any part of the official
solution.
- Avoid over-fragmenting: do not split a routine algebra manipulation
into 3+ separate 1-pt bullets.
- Keep notation consistent with the official solution; if you
introduce new symbols, define them.

______________________________________________________________

Here is an example problem from a past math competition, along with
human-written solutions and the corresponding marking scheme.
Please study the problem and solution carefully, and use them to
understand how the marking scheme was constructed.

### Problem
A positive integer n is called beautiful if, for every integer 4 <= b
<= 10000, the base-b
representation of n contains the consecutive digits 2, 0, 2, 3 (in
this order, from left to right).
Determine whether the set of all beautiful integers is finite.

### Solution
\[
N_{4}<N_{5}<N_{6}<\ldots
\]

such that for every \(k=4,5, \ldots\), the number \(N_{k}\) contains
\(2023_{b}\) in every base \(4 \leq b \leq k\). This will solve the
problem because \(N_{10000}, N_{10001}, \ldots\) will be the requested
infinite set.

For the base case, take \(N_{4}=2023_{4}\).\\
For the inductive step, here is one of many valid recipes. We are
going to select
\[
N_{k}=N_{k-1}+c \cdot(k \ell)^{e}
\]
where the ingredients \(c, \ell, e\) are selected to satisfy:

\begin{itemize}
  \item \(\ell\) is the product of all primes at most \(k\) which are
  relatively prime to \(k\) (in particular,
```

```
  \(\operatorname{gcd}(k, \ell)=1\) );
  \item \(e\) is large enough that for each \(b=4,5, \ldots, k\),
  the largest power of \(b\) dividing \((k \ell)^{e}\) is greater than
  \(b \cdot N_{k-1}\);
  \item \(c\) is chosen to satisfy the modular congruence
\end{itemize}

\[
c \cdot \ell^{e} \equiv 2 k^{3}+0 k^{2}+2 k+3 \quad
\left(\bmod k^{4}\right)
\]

which is possible since \(\operatorname{gcd}\left(k^{4},
\ell^{e}\right)=1\).\\
With these ingredients, for all the smaller bases \(4,5,
\ldots, k-1\), the ending of \(N_{k}\) in base- \(b\) is the same
as in \(N_{k-1}\) (since \((k \ell)^{e}\) is a multiple of a large
enough power of \(b\) ). On the other hand, we've embedded \(2023_{k}\)
into the base- \(k\) representation of \(N_{k-1}\), because the
coefficients of \(k^{e+3}, k^{e+2}, k^{e+1}, k^{e}\) in the
base-\(k\) representation are exactly \(2,0,2,3\).

### Marking Scheme
1. Checkpoints (max 7 pts total)
Merely claiming there is some congruence of some sort is not sufficient
to pass the benchmark. For solutions that fail to achieve this
benchmark, the following partial credits are available but not additive:
- 1 point for showing that the existence of a single beautiful number
implies the existence of infinitely many, e.g. by adding large powers
of 2023!.
- 1 point for a serious induction attempt or construction but which
botches the main difficulty mentioned above.
- 2 points for a solution that additionally has the idea in the
indirect construction solution of ensuring compatibility by
picking a certain integer parameter $t_i \in \{0, 1, \cdots, b_k - 1\}$
in a consecutive range for a sufficiently large $k$.
- 7 points if they are completely correct.

2. Zero-credit items
- 0 points for yes/no answer alone.
- 0 points for just mentioning induction.
- 0 points for base cases like b = 4.
- 0 points for a solution that only works on coprime moduli. This
includes, e.g. showing there is a number which has 2023b for every
base 4 <= b <= 2023 which is the power of a prime (that is,
$b \in \{4, 5, 7, 8, 9, 11, 13, . . . , 2011\}$).

3. Deductions
- -1 point for a minor error such as:
 - completely omitting the base case of the induction;
 { using an integer parameter which is not large enough as stated but
    could easily be changed to be large enough.
- -2 points for a more serious but non-central flaw in one of the
steps of an inductive approach.

_______________________________________________________

Below is the problem with its correct solution you need to read and
generate the marking scheme for.

### PROBLEM
<insert problem statement>

### OFFICIAL SOLUTION
```

```
<insert full solution>
```

## K    EVALUATOR PROMPTS

Below we list all prompts used in our evaluation. Each prompt is shown verbatim.

---

**With Reference Solution and Marking Scheme**

You are an **expert math proof grader**. You are judging the
correctness of an LLM-generated proof for a math problem.

### Input

Your input will consist of:

* **Problem Statement**: A mathematical problem that the proof
is attempting to solve.
* **Reference Solution**: A correct solution or proof provided
for reference. This is **not necessarily the only valid solution**.
If the problem requires a final numeric or algebraic answer, this
section contains the correct answer, which should be the only
accepted final answer (though alternative reasoning paths are valid).
* **Marking Scheme**: A problem-specific grading rubric (0{7 scale)
with checkpoints, zero-credit items, and deductions. **Treat this
scheme as advisory guidance, not a script.** Use it to anchor
scoring, but **do not require** the proof to follow the same
order, lemmas, or technique if its reasoning is mathematically
sound.
* **Proof Solution**: The proof that you need to evaluate. This
proof may contain errors, omissions, or unclear steps. The proof
was generated by another language model.

### Task

Analyze the proof carefully.

**Core principles (in order of precedence):**
1) **Mathematical validity** of the proof's reasoning and conclusion.
2) **Problem constraints** (e.g., unique required final value;
forbidden tools if stated).
3) **Advisory mapping to the marking scheme** (checkpoints/
deductions), allowing different orders and techniques.
4) **Reference solution** as an anchor for sufficiency, not
exclusivity.

**Alternative-approach policy:**
- If the proof uses a different but valid method, **map its
steps to equivalent rubric checkpoints** (same logical role)
and award points accordingly.
- **Do not penalize** solely for re-ordering steps, using
different lemmas, or giving a correct shortcut, **unless**
the problem forbids it.
- Apply zero-credit items/deductions **only when the underlying
issue actually occurs** in the given proof's approach; **do not
auto-penalize** for omitting a rubric step that is unnecessary
under the alternative method.
- Avoid double-counting mutually exclusive items; if two items
solve the same logical gap, **award the larger only**.
- If the final numeric/algebraic answer is wrong where uniqueness
is required, award only partial credit justified by correct
intermediate reasoning.

**Rigor and evidence:**
- Award credit for intermediate claims **only if adequately justified** within the proof (not merely asserted).
- If a step is plausible but under-justified, award **conservative partial credit** and note what is missing.

**What to produce:**
- Identify logical errors, incorrect steps, or unclear reasoning.
- Give a **score between 0 and 7** with a **detailed assessment**.
- **Within the assessment text**, show clearly how the score was derived:
  - Which rubric checkpoints (or their **mapped equivalents**) were earned and the points you awarded.
  - Any zero-credit items or deductions you applied (and why).
  - How these add up to the final integer score in [0-7].

### Output Format

Respond with **only** well-formed XML using the structure below. Do not include any extra text or Markdown.

**Requirements:**
- `<score>` must be an integer in [0, 7].
- `<assessment>` must be a **detailed analysis** that explains your reasoning step-by-step and provides a clear **rationale for the score**. Reference specific claims/lines if present. Include the scoring breakdown **in prose** here (earned checkpoints or mapped equivalents, deductions, and subtotal → final score).
- `<errors>` must be a list of specific issues (empty if score = 7).

Example output:

```
<score>0</score>
<assessment>The proof shows a good understanding of the main idea,
but has some unclear reasoning and minor mistakes...</assessment>
<errors>
  1. specific error 1,
  2. specific error 2,
  ...
</errors>
```

---------------------------------------------------------
**Problem Statement**
{problem}

**Reference Solution**
{human_solution}

**Marking Scheme**
{marking_scheme}

**Proof Solution**
{solution}

---

## Basic Evaluation Template

You are an **expert math proof grader**. You are judging the correctness of an LLM-generated proof for a math problem.

### Input

Your input will consist of:

* **Problem Statement**: A mathematical problem that the proof is attempting to solve.
* **Proof Solution**: The proof that you need to evaluate. This proof may contain errors, omissions, or unclear steps. The proof was generated by another language model.

### Task

Analyze the proof carefully.

* Identify logical errors, incorrect steps, or unclear reasoning.
* Give an **integer** score between 0 and 7 with a brief overall assessment.

### Output Format

Respond with **only** well-formed XML using the structure below. Do not include any extra text or Markdown.

**Requirements:**
- `<score>` must be an integer in [0, 7].
- `<assessment>` must be a **detailed analysis** that explains your reasoning step-by-step and provides a clear **rationale for the score**. Reference specific claims/lines if present.
- `<errors>` must be a list of specific issues (empty if score = 7).

Example output:

<score>0</score>
<assessment>The proof shows a good understanding of the main idea, but has some unclear reasoning and minor mistakes...</assessment>
<errors>
  1. specific error 1,
  2. specific error 2,
  ...
</errors>

### Scoring Guidelines (0-7 scale)

* **0**: Completely incorrect; proof is irrelevant, nonsensical, or shows no understanding.
* **1-2**: Very poor; major logical flaws, does not solve the problem, but may contain fragments of relevant reasoning.
* **3-4**: Partial progress; captures some correct reasoning or key ideas, but has significant logical errors, missing steps, or incomplete arguments that make the proof invalid overall.
* **5-6**: Largely correct; the proof is overall valid and reaches the correct conclusion. Contains only **minor issues** (e.g., small calculation mistakes, notation slips, or slightly unclear wording) that do not undermine correctness.
* **7**: Fully correct; the proof is complete, logically sound, and clearly presented with no substantive errors.

---------------------------------------------------------
**Problem Statement**
{problem}

**Proof Solution**
{solution}

**With Reference Solution**

You are an **expert math proof grader**. You are judging the correctness of an LLM-generated proof for a math problem.

### Input

Your input will consist of:

* **Problem Statement**: A mathematical problem that the proof is attempting to solve.
* **Reference Solution**: A correct solution or proof provided for reference. This is **not necessarily the only valid solution**. If the problem requires a final numeric or algebraic answer, this section contains the correct answer, which should be the only accepted final answer (though alternative reasoning paths are valid).
* **Proof Solution**: The proof that you need to evaluate. This proof may contain errors, omissions, or unclear steps. The proof was generated by another language model.

### Task

Analyze the proof carefully.

* Compare the proof against the reference solution where relevant.
* Identify logical errors, incorrect steps, or unclear reasoning.
* Give a score between 0 and 7 with a brief overall assessment.

### Output Format

Respond with **only** well-formed XML using the structure below. Do not include any extra text or Markdown.

**Requirements:**
- `<score>` must be an integer in [0, 7].
- `<assessment>` must be a **detailed analysis** that explains your reasoning step-by-step and provides a clear **rationale for the score**. Reference specific claims/lines if present.
- `<errors>` must be a list of specific issues (empty if score = 7).

Example output:

<score>0</score>
<assessment>The proof shows a good understanding of the main idea but has some unclear reasoning and minor mistakes...</assessment>
<errors>
  1. specific error 1,
  2. specific error 2,
  ...
</errors>

### Scoring Guidelines (0-7 scale)

* **0**: Completely incorrect; proof is irrelevant, nonsensical, or shows no understanding.
* **1-2**: Very poor; major logical flaws, does not solve the problem, but may contain fragments of relevant reasoning.
* **3-4**: Partial progress; captures some correct reasoning or key ideas, but has logical errors, missing steps, or incomplete arguments that make the proof invalid overall.
* **5-6**: Largely correct; the proof is overall valid and reaches the correct conclusion. Contains only **minor issues** (e.g., small calculation mistakes, notation slips, or slightly unclear wording) that do not undermine correctness.

* **7**: Fully correct; the proof is complete, logically sound, and clearly presented with no substantive errors.

----------------------------------------------------------
**Problem Statement**
{problem}

**Reference Solution**
{human_solution}

**Proof Solution**
{solution}

---

**With Reference Solution and Marking Scheme (Strict)**

You are an **expert math proof grader**. You are judging the correctness of an LLM-generated proof for a math problem.

### Input

Your input will consist of:

* **Problem Statement**: A mathematical problem that the proof is attempting to solve.
* **Reference Solution**: A correct solution or proof provided for reference. This is **not necessarily the only valid solution**. If the problem requires a final numeric or algebraic answer, this section contains the correct answer, which should be the only accepted final answer (though alternative reasoning paths are valid).
 * **Marking Scheme**: A problem-specific grading rubric (0-7 scale) with checkpoints, zero-credit items, and deductions. You must follow this scheme when assigning points.
* **Proof Solution**: The proof that you need to evaluate. This proof may contain errors, omissions, or unclear steps. The proof was generated by another language model.

### Task

Analyze the proof carefully.

* Compare the proof against the reference solution and the marking scheme.
* Award points according to the marking scheme's checkpoints, zero-credit items, and deductions.
* Identify logical errors, incorrect steps, or unclear reasoning.
* Give a score between 0 and 7 with a brief overall assessment.
* Show clearly how the score was derived:
  * Which checkpoints were earned (with awarded points).
  * Any zero-credit items or deductions applied.
  * How the subtotal leads to the final score (0-7).

### Output Format

Respond with **only** well-formed XML using the structure below. Do not include any extra text or Markdown.

**Requirements:**
- `<score>` must be an integer in [0, 7].
- `<assessment>` must be a **detailed analysis** that explains your reasoning step-by-step and provides a clear **rationale for the score**. Reference specific claims/lines if present.
- `<errors>` must be a list of specific issues (empty if score = 7).

```
Example output:

<score>0</score>
<assessment>The proof shows a good understanding of the main idea
but has some unclear reasoning and minor mistakes...</assessment>
<errors>
  1. specific error 1,
  2. specific error 2,
  ...
</errors>

------------------------------------------------------------
**Problem Statement**
{problem}

**Reference Solution**
{human_solution}

**Marking Scheme**
{marking_scheme}

**Proof Solution**
{solution}
```

## With Reference Solution and Marking Scheme (most basic)

```
You are an expert grader for math proofs. Judge the proof's
mathematical correctness based on the reference solution and
the marking scheme, return an integer score between 0 and 7.

INPUTS:
- Problem Statement
- Reference Solution (correct but not exclusive)
- Marking Scheme (0-7) with checkpoints and deductions | use
as guidance, not a script
- Proof Solution (from an LLM)

OUTPUT (XML only; no extra text):
<score>[integer 0-7]</score>
<assessment>[step-by-step rationale with scoring breakdown in prose]
</assessment>
<errors>[numbered list of specific issues; empty if none]</errors>

------------------------------------------------------------

**Problem Statement**
{problem}

**Reference Solution**
{human_solution}

**Marking Scheme**
{marking_scheme}

**Proof Solution**
{solution}
```

## With Reference Solution and Marking Scheme (basic)

```
You are an expert grader for math proofs.

INPUTS:
```

```
- Problem Statement
- Reference Solution (correct but not exclusive)
- Marking Scheme (0-7) with checkpoints and deductions  use as
guidance, not a script
- Proof Solution (from an LLM)

TASK:
Judge the proof's mathematical correctness. Prefer validity > problem
constraints > marking scheme alignment > reference solution. If the
proof uses a different valid method, map its steps to equivalent
marking scheme checkpoints and award points. If a unique final answer
is wrong, give partial credit only for justified intermediate
reasoning.

OUTPUT (XML only; no extra text):
<score>[integer 0-7]</score>
<assessment>[step-by-step rationale with scoring breakdown in prose]
</assessment>
<errors>[numbered list of specific issues; empty if none]</errors>

----------------------------------------------------------

**Problem Statement**
{problem}

**Reference Solution**
{human_solution}

**Marking Scheme**
{marking_scheme}

**Proof Solution**
{solution}
```

### With Marking Scheme (no reference solution)

```
You are an **expert math proof grader**. You are judging the
correctness of an LLM-generated proof for a math problem.

### Input

Your input will consist of:

* **Problem Statement**: A mathematical problem that the proof is
attempting to solve.
* **Marking Scheme**: A problem-specific grading rubric (0-7 scale)
with checkpoints, zero-credit items, and deductions. You must follow
this scheme when assigning points.
* **Proof Solution**: The proof that you need to evaluate. This proof
may contain errors, omissions, or unclear steps. The proof was
generated by another language model.

### Task

Analyze the proof carefully.

* Follow the marking scheme exactly: award checkpoints, apply
zero-credit items, and apply any deductions/caps as specified.
* Identify logical errors, incorrect steps, or unclear reasoning.
* Give a score between 0 and 7 with a brief overall assessment.
* Show clearly how the score was derived:
  * Which checkpoints were earned (with awarded points).
  * Any zero-credit items or deductions applied.
```

```
   * How the subtotal leads to the final score (0-7).

### Output Format

Respond with **only** well-formed XML using the structure below.
Do not include any extra text or Markdown.

**Requirements:**
- `<score>` must be an integer in [0, 7].
- `<assessment>` must be a **detailed analysis** that explains your
reasoning step-by-step and provides a clear **rationale for the
score**. Reference specific claims/lines if present.
- `<errors>` must be a list of specific issues (empty if score = 7).

Example output:

<score>0</score>
<assessment>The proof shows a good understanding of the main idea,
but has some unclear reasoning and minor mistakes...</assessment>
<errors>
  1. specific error 1,
  2. specific error 2,
  ...
</errors>

---------------------------------------------------------
**Problem Statement**
{problem}

**Marking Scheme**
{marking_scheme}

**Proof Solution**
{solution}
```

**With Reference Solution and Marking Scheme (more detailed)**

```
You are an **expert math proof grader**. You are judging the
correctness of an LLM-generated proof for a math problem.

### Input

Your input will consist of:

* **Problem Statement**: A mathematical problem that the proof is
attempting to solve.
* **Reference Solution**: A correct solution or proof provided for
reference. This is **not necessarily the only valid solution**. If
the problem requires a final numeric or algebraic answer, this section
contains the correct answer, which should be the only accepted final
answer (though alternative reasoning paths are valid).
* **Marking Scheme**: A problem-specific grading rubric (0-7 scale)
with checkpoints, zero-credit items, and deductions. You must follow
this scheme when assigning points.
* **Proof Solution**: The proof that you need to evaluate. This proof
may contain errors, omissions, or unclear steps. The proof was
generated by another language model.

### How to Use the Marking Scheme (mandatory)

1. Checkpoints parsing & awarding
- Treat each checkpoint exactly as written. Respect its tag:
  - [additive]: award all applicable items in that bullet/group.
```

    – [max k]: award up to k points from the items in that
    bullet/group (choose the best-matching ones; do not exceed k).
- If items are nested with \award the larger only", and more than
one applies, award only the larger point value.
 – If the scheme presents parallel checkpoint chains (alternative
 legitimate paths), score the single chain or combination that yields
 the highest valid total without violating exclusivity or [max k]
 caps. Do not double-count equivalent steps across mutually
 exclusive paths.
– If a catch-all checkpoint is provided for a fully correct
alternative proof using the same underlying idea, you may award up
to its stated maximum only when the student's argument is complete
and logically valid for that idea.

2. Zero-credit items

If the proof relies on any listed zero-credit arguments, award 0
for those parts. Do not add points for restatements, conjectures
without proof (especially in geometry), or dead-ends.

3. Deductions (apply at most one)

– Identify applicable deductions and apply only the single largest
(e.g., -1, -2, or cap at x/7).
– Apply a cap by truncating the post-checkpoint subtotal to x before
finalizing the score.
– Never reduce the score below 0. Cosmetic slips (notation, arithmetic,
wording) do not trigger deductions unless they break validity.

4. Final answer consistency (when applicable)

If the reference solution gives a definitive final answer, the
candidate solution's final answer must be **correct/equivalent**.
If not, follow the marking scheme's checkpoints/deductions; typically,
a wrong final answer prevents awarding the \conclusion" checkpoint.

5. Arithmetic & bounds
– Checkpoint awards are integers. Subtotal <= 7 by construction.
– After applying the single largest deduction/cap, the final score
is an integer in [0, 7].

### Task

Analyze the proof carefully.

* Compare the proof against the reference solution and the marking
scheme.
* Award points according to the marking scheme's checkpoints,
zero-credit items, and deductions.
* Identify logical errors, incorrect steps, or unclear reasoning.
* Give a score between 0 and 7 with a brief overall assessment.
* Show clearly how the score was derived:
* Which checkpoints were earned (with awarded points).
* Any zero-credit items or deductions applied.
* How the subtotal leads to the final score (0-7).

### Output Format

Respond with **only** well-formed XML using the structure below.
Do not include any extra text or Markdown.

**Requirements:**

```
- `<score>` must be an integer in [0, 7].
- `<assessment>` must be a **detailed analysis** that explains your
reasoning step-by-step and provides a clear **rationale for the
score**. Reference specific claims/lines if present.
- `<errors>` must be a list of specific issues (empty if score = 7).

Example output:

<score>0</score>
<assessment>The proof shows a good understanding of the main idea,
but has some unclear reasoning and minor mistakes...</assessment>
<errors>
  1. specific error 1,
  2. specific error 2,
  ...
</errors>

--------------------------------------------------------
**Problem Statement**
{problem}

**Reference Solution**
{human_solution}

**Marking Scheme**
{marking_scheme}

**Proof Solution**
{solution}
```

