# OpenReview forum: "Reliable Fine-Grained Evaluation of Natural Language Math Proofs"
_ICLR.cc/2026/Conference — ICLR 2026 Poster_

### Official Review · Reviewer_EDhE · 2025-10-20

**Soundness:** 2
**Presentation:** 2
**Contribution:** 3
**Rating:** 4
**Confidence:** 5

**Summary:**

This paper evaluates the feasibility and effectiveness of LLMs as judges for the continuous-scale grading of natural language solutions to mathematical problems. The authors present a pipeline for 1) automatically generating grading rubrics, 2) manually verifying LLM-generated solutions, and 3) investigating various aspects of an LLM-as-a-judge system. This process resulted in the `ProofBench` dataset of 393 expertly graded solutions, used for evaluating the backbone model and prompting style of an LLM judge. The work also presents `ProofGrader`: the best-performing combination of a judgment ensemble using the o3 model, with a marking scheme and potential solutions as additional reference information. As a proof of concept for `ProofGrader`'s utility, the authors show that the system can be used as a solution selector in a best-of-8 setting, achieving a score that approaches the human oracle's performance.

**Strengths:**

1. The topic of the paper is clearly well-motivated, relevant, timely, impactful, and provides opportunities for further developments in the field
2. The paper is overall well-written and conveys the high-level idea well.
3. The authors have investigated several factors affecting the performance of an LLM-as-a-judge, identifying the backbone model and a well-defined marking scheme as the most significant factors for its performance. This provides readers with clear guidance on the information necessary for achieving accurate automatic grading.
4. This work provides the currently largest dataset of expert-annotated solutions graded on a continuous scale, covering a wide range of challenging recent problems suitable for further LLM-as-a-judge validation.
5. The best evaluated settings for the LLM-as-a-judge are shown to also correlate with a better performance as a best-of-n selector. This is a very desirable result, showing that the comparisons performed in the analysis are valid on downstream tasks.
6. The authors provide the prompts necessary for running the LLM-as-a-judge, aiding reproducibility.
7. The annotator calibration phase is a valuable step for minimizing error and noise in the final dataset.

**Weaknesses:**

For this section, I have labelled each comments as either:
 - *Critical*: These weaknesses have significantly impacted the score, and addressing/not addressing them could positively or negatively affect my final recommendation.
- *Important*: Addressing these weaknesses would, in my opinion, significantly strengthen the paper's quality and/or clarity.
- *Minor*: These are nitpicks that, while valuable to address, have not impacted the score of this assessment.

1. Reproducibility
    - (Critical) The authors claim `ProofBench` as one of their core contributions. However, despite promising to open-source the dataset upon publication, I see no reason why this was not done as part of the supplementary material. Not providing a core contribution of the paper for peer review significantly hinders the process. Upon release, the authors should also ensure they include an appropriate license, given the nature of the source materials.
    - (Important) The annotation pipeline, described in A.3, is missing a substantial amount of details. Further elaboration can be found in the **Questions** section.

2. Methodology
    - (Critical) The authors have not reported the reliability of their human grading. They describe having performed double-grading on 20% of the solutions but have not reported any inter-annotator agreement statistics. In particular, they mention that they "adjudicate all flagged disagreements," however, it is not clear what threshold for a disagreement constitutes a flag, how often flags occurred, or how they resolved these inconsistencies. The perceived reliability of the remaining 80% of grades is highly dependent on clarifying this process.
    - (Critical) Generating rubrics is an incredibly challenging task in practice. This paper involves an automated system that is difficult to verify without significant manual intervention. Details about how this process was refined are lacking. Furthermore, no discussion is presented on how the annotators verified that these rubrics adhered to a standard for high-quality marking schemes. For example, the rubric in Appendix A.6 gives a total of 3 points for initial observations that seem relatively trivial compared to the rest of the proof. While my interpretation of this rubric is a personal opinion, I believe the authors should clarify how they ensured that the rubrics adhere to a high standard of quality, reflecting authentic evaluation practices.
    - (Important) The authors claim that in Section 6 they investigate the judging framework as a reward model. However, the described setting appears to lack a clear, realistic application. In particular:

        * If used within a solver's selection system, as described in the paper, such applications often aim to solve a problem without access to a reference solution. In that case, the best result seems to be `o3`, at around 3 average points, which is considerably lower than the human oracle's 4.21.
        * If used as a reward model for training, as proposed in Sections 1 and 7, the presented setups are all too expensive to be scalable to a full RL (or other) pipeline.

      The authors should clarify what a potential use case for this result can be.
    - (Minor) The paper makes no distinction between results on undergraduate- and high-school-level problems, and whether this is a relevant factor.

3. Dataset
    - (Important) The dataset consists of problems from 2022-2025 from popular competitions (IMO, USAMO, Putnam). The earlier years precede the knowledge cutoff dates for the models tested. The authors should discuss whether test-set contamination has impacted the results and, if so, how significant the effect is.
    - (Important) The authors claim to have sourced their problems and solutions from official sources. However, to the best of my knowledge, the USAMO and USATST do not publicly release their competition materials. The authors should clarify this point in their rebuttal.

4. Writing and Clarity
    - (Minor) The work never explicitly states which configuration constitutes `ProofGrader`. While it can be inferred from the results, it would be best if the authors defined this in the earlier sections of the paper.
    - (Minor) Most models presented in the paper are not cited according to best practices. Standard practice is to cite the official model cards from the providers, if one exist.
    - (Minor) In Section 5.3, the authors refer to an "ensemble" of evaluators. This term is usually reserved for a collection of **different** models or algorithms applied to the same task, rather than for multiple samples from a single algorithm.

**Questions:**

1. Can the authors address all the concerns listed in the **Weaknesses** section?
2. The authors describe the marking scheme generation methodology in Section 3.1 and later in Appendix A.3. However, the description lacks sufficient detail. Can the authors clarify:
    - The model families they tested for rubric generation?
    - What the prompts (with and without examples) entailed?
    - How the annotators interacted with the initially generated rubrics, e.g., what were their instructions, how was consensus achieved, and what was the extent of disagreement prior to adjudication?
    - What was the average rubric quality rating in the final iteration?
    - How was the best configuration selected, was it based solely on the average score?
3. The authors have measured the bias of the graders with respect to the human grades. However, when breaking down by model, [1] and [2] show a clear positive bias toward a model's own solutions. Can the authors report similar metrics and discuss the implications?
4. In what realistic setting can the authors' framework from Section 6 be applied (refer to comments in W2.3)?
5. When evaluating different provers, the authors have constrained themselves to using a score-based selection system. How does this compare to using a tournament-style approach?
6. The best-of-n evaluation was done on 29 selected problems. How were these problems selected and what was their difficulty distribution?
7. The authors claim in 3.2.3 that Staged Evaluation is "particularly effective for improving the performance of weaker backbone models." This is a very strong claim, given that this observation is only seen for the `o4-mini` model. Can the authors consider running additional experiments to support this statement or temper the claim to reflect the limited evidence?
8. For 2/3 models in 5.2.2, the *Strict* setting yields more accurate judgement than the *Norm* one. Do the authors have any qualitative explanations for this?

## Current rating

I have given this paper a score of **4: Borderline Reject**. The contribution is valid, important, and potentially impactful to the field. However, the lack of transparency on some aspects, particularly reproducibility, prevents me from assigning a higher score. I would be happy to raise my score if the authors address the majority of my concerns during the rebuttal and discussion period.

### References

[1] Dekoninck et al. The open proof corpus: A large-scale study of llm-generated mathematical proofs. arXiv preprint arXiv:2506.21621, 2025.

[2] Petrov et al. Proof or bluff? evaluating llms on 2025 usa math olympiad. arXiv preprint arXiv:2503.21934, 2025.

**Details Of Ethics Concerns:**

None.

---

> ### Author Response · Authors · 2025-11-20
> **Response to Reviewer EDhE (1/n)**
>
> We thank Reviewer EDhE for the detailed and constructive feedback! Your suggestions have helped us improve the paper a lot.
>
> We respectfully encourage the reviewer to review our revised paper, where many concerns have been addressed through updates to the text and experiments. Below, we also address each concern in the order presented in your review, with references to the corresponding sections in the revised paper.
>
> ### 1. Reproducibility
>
> > **1. (Critical) The authors claim ProofBench as one of their core contributions. However, despite promising to open-source the dataset upon publication, I see no reason why this was not done as part of the supplementary material. Not providing a core contribution of the paper for peer review significantly hinders the process. Upon release, the authors should also ensure they include an appropriate license, given the nature of the source materials.**
>
> We have now provided our datasets via an anonymous repository (https://anonymous.4open.science/r/ProofBench-Supplementary-4DD2). The dataset includes all annotated solutions, marking schemes, metadata as described in the paper.
>
> Regarding licensing: ProofBench contains problems sourced from publicly available mathematical competitions (e.g., IMO, APMO). We provide clear attribution for all such problems. Upon camera-ready release, we will publish the dataset under an appropriate open license (e.g., CC BY 4.0) together with explicit attribution and usage guidelines to respect the policies of the original competition sources.
>
> We greatly appreciate your feedback and welcome any suggestions regarding the dataset or licensing approach during the review process.
>
> > **2. (Important) The annotation pipeline, described in A.3, is missing a substantial amount of details. Further elaboration can be found in the Questions section.**
>
> This is an important concern! We have included more details in A.3 in the revised paper. Please also see our answers to the Questions section.
>
> ---
> ### 2. Methodology
>
> > **1. (Critical) The authors have not reported the reliability of their human grading. They describe having performed double-grading on 20% of the solutions but have not reported any inter-annotator agreement statistics. In particular, they mention that they "adjudicate all flagged disagreements," however, it is not clear what threshold for a disagreement constitutes a flag, how often flags occurred, or how they resolved these inconsistencies. The perceived reliability of the remaining 80% of grades is highly dependent on clarifying this process.**
>
> Please refer to our global response for the updates to our dataset and question #1. We have more than 40% double-graded data in both datasets used in sections 3 and 4.
>
> > **2. (Critical) Generating rubrics is an incredibly challenging task in practice. This paper involves an automated system that is difficult to verify without significant manual intervention. Details about how this process was refined are lacking. Furthermore, no discussion is presented on how the annotators verified that these rubrics adhered to a standard for high-quality marking schemes. For example, the rubric in Appendix A.6 gives a total of 3 points for initial observations that seem relatively trivial compared to the rest of the proof. While my interpretation of this rubric is a personal opinion, I believe the authors should clarify how they ensured that the rubrics adhere to a high standard of quality, reflecting authentic evaluation practices.**
>
> We thank the reviewer for highlighting this concern. We fully agree that generating high-quality marking schemes is an inherently challenging task. In the revised paper, we have expanded Section A.3 to describe the development and refinement process in greater detail. Our approach is a practical solution when no official or existing marking schemes are available, which is the case for the competitions we considered.
>
> To ensure quality, expert annotators were instructed to carefully review each generated scheme and to consult with one another whenever they identified potential issues. They were also advised to treat the marking schemes as flexible references rather than rigid checklists. This process helped maintain both consistency and critical oversight.
>
> Regarding the specific example mentioned, we acknowledge that the issue raised is valid and was similarly noted by our annotators (see Appendix A.3). While this particular rubric is not representative of the overall quality, such imperfections are sometimes unavoidable when the generator overlooks non-trivial reasoning steps. However, we think that complementary components of the rubric, such as the zero-point sections and our grading instructions, help mitigate this limitation and preserve the fairness and reliability of evaluation.

---

> > ### Author Response · Authors · 2025-11-20
> > **Response to Reviewer EDhE (2/n)**
> >
> > > **3. (Important) The authors claim that in Section 6 they investigate the judging framework as a reward model. However, the described setting appears to lack a clear, realistic application. In particular: If used within a solver's selection system, as described in the paper, such applications often aim to solve a problem without access to a reference solution. In that case, the best result seems to be o3, at around 3 average points, which is considerably lower than the human oracle's 4.21; If used as a reward model for training, as proposed in Sections 1 and 7, the presented setups are all too expensive to be scalable to a full RL (or other) pipeline. The authors should clarify what a potential use case for this result can be.**
> >
> > Thank you for raising this! In fact, our intent of best-of-n experiments is not to present a new test-time strategy or a drop-in RL reward model, but to use the “best-of-n selection” setup as a proxy task to evaluate an evaluator’s ability to rank and select higher-quality proofs. Concretely, §6 compares scoring scales and checks consistency with §5 to ask: does a finer-grained, marking-scheme-based evaluator actually surface better solutions?
> >
> > * **On scalability as a reward model.** We agree that running the full multi-pass ProofGrader inside an RL loop is expensive. We position our results as design guidance for reward modeling: they demonstrate that fine-grained scoring and marking-scheme matter for detecting non-trivial progress and high quality proofs.To our knowledge, no prior work has demonstrated such findings. Practically, this gives two feasible uses:
> >    * **Offline synthetic data curation.** Use ProofGrader offline to score/filter large candidate corpora and train a cheaper distilled or RL’ed reward model that is then used online. ProofGrader can also be used for rejection sampling and correctness filtering during distillation.
> >    * **Feasibility studies.** A single-pass variant is markedly cheaper and suitable for small-scale RL, before distilling to lighter reward models. Despite its latency, evaluator time is often much smaller than rollout generation, which can take tens of minutes per step on local hardware. The financial cost is also reasonable when training SOTA frontier models.
> > * **On the “no reference at test time” concern.** We agree that evaluators with access to reference solutions are not allowed at test time for a solver. Our revised paper clarifies that these settings are for analysis of evaluator signal, not as a proposed deployment recipe. We want to leave it for future work to design better reference-free evaluators.
> >
> > > **4. (Minor) The paper makes no distinction between results on undergraduate- and high-school-level problems, and whether this is a relevant factor.**
> >
> > In our dataset, Putnam represents undergraduate-level problems, while the remaining competitions correspond to high-school level.
> >
> > For generator performance, Figure 2(f) in the revised paper shows that Putnam problems are, on average, easier for generators (they yield the highest mean scores among all competitions).
> >
> > For evaluator performance, we observe only minor differences between the two levels: ProofGrader attains 0.912 / 1.548 MAE/RMSE on undergraduate problems and 0.931 / 1.511 on high-school problems. Overall, the distinction has limited impact on evaluator performance.

---

> > > ### Author Response · Authors · 2025-11-20
> > > **Response to Reviewer EDhE (3/n)**
> > >
> > > ### 3. Dataset
> > >
> > > > **1. (Important) The dataset consists of problems from 2022-2025 from popular competitions (IMO, USAMO, Putnam). The earlier years precede the knowledge cutoff dates for the models tested. The authors should discuss whether test-set contamination has impacted the results and, if so, how significant the effect is.**
> > >
> > > Thank you for raising this important concern. We first provide the analysis of whether contamination could inflate generation performance on problems that appear in a model’s pretraining data. To quantify this, we will report ablations split by knowledge-cutoff and discuss any observed differences. We use the following cutoffs (with sources):
> > >
> > > * o3: 2024-06-01 (https://platform.openai.com/docs/models/o3)
> > > * Gemini-2.5-Pro: 2025-01 (https://deepmind.google/models/gemini/pro/)
> > > * DeepSeek-R1: 2024-07 (https://explodingtopics.com/blog/list-of-llms) (The newer version should share the same cutoff)
> > >
> > > Here we present generator performance before vs. after each model’s cutoff:
> > > | Model         | Before Cutoff (Contaminated) Count | Before Cutoff (Contaminated) Mean Grading | After Cutoff (Clean) Count | After Cutoff (Clean) Mean grading | Difference |
> > > |--------------|-----------------------------------|------------------------------------------|---------------------------|----------------------------------|-----------|
> > > | OpenAI o3    | 92                                 | 3.022                                     | 53                         | 2.585                             | -0.437     |
> > > | Gemini-2.5-Pro | 116                              | 2.733                                     | 29                         | 1.724                             | -1.009     |
> > > | DeepSeek-R1  | 98                                 | 1.949                                     | 47                         | 1.553                             | -0.396     |
> > >
> > > Our results provide evidence of data contamination, which strengthens the case for prioritizing fresh-data evaluations on math benchmarks. Note that absolute rates are not directly comparable across models because the problem counts and knowledge cutoffs differ. We also observe variation across competitions; detailed per-competition analyses are included in Appendix A.4 of the revised paper.
> > >
> > > We additionally report per-year splits for ProofGrader on ProofBench using o3’s cutoff.
> > >
> > > | Year | After Cutoff |   MAE |  RMSE | WTA(<=1) |  Bias  |
> > > |------|--------------|------|------|--------|-------|
> > > | 2022 | No           | 0.858 | 1.351 |   79.4% | -0.132 |
> > > | 2023 | No           | 0.925 | 1.635 |   76.3% |  0.013 |
> > > | 2024 | Yes          | 0.898 | 1.484 |   78.9% | -0.053 |
> > > | 2025 | Yes          | 1.046 | 1.601 |   75.9% |  0.253 |
> > >
> > > These results show that contamination has minimal impact on evaluator performance: performance on 2022–2023 (pre-cutoff) and 2024–2025 (post-cutoff) is similar, suggesting that ProofGrader’s behavior is robust to potential contamination in the underlying problems/solutions.
> > >
> > > > **2. (Important) The authors claim to have sourced their problems and solutions from official sources. However, to the best of my knowledge, the USAMO and USATST do not publicly release their competition materials. The authors should clarify this point in their rebuttal.**
> > >
> > > Thank you for pointing this out. We apologize for the confusion in our initial submission. The USAMO and USA TST problems and solutions are not released on their official websites, and at the time of submission, the IMO 2025 solutions were also not yet available (though problems and other years were). For these portions of our dataset (USAMO, USA TST, and IMO 2025 solutions), we sourced the materials from Evan Chen’s website (https://web.evanchen.cc/problems.html), which we consider a highly reliable source. We would appreciate any suggestions for alternative sources that could further improve our dataset.
> > > We have clarified these details in the revised paper, specifically in Section 2 and Appendix A.2.
> > >
> > > ---
> > > ### 4. Writing and Clarity
> > >
> > > > **1. (Minor) The work never explicitly states which configuration constitutes ProofGrader. While it can be inferred from the results, it would be best if the authors defined this in the earlier sections of the paper.**
> > >
> > > Yes, sorry for the confusion. The ProofGrader means an evaluator that combines a strong reasoning backbone LM (o3), rich context from reference solutions and marking schemes, and a simple ensembling method. We have added a paragraph for clarification at the end of section 3.4.
> > >
> > > > **2. (Minor) Most models presented in the paper are not cited according to best practices. Standard practice is to cite the official model cards from the providers, if one exists.**
> > >
> > > Thank you for pointing this out! We added correct citations in the revised version; please see section 2 and section 3.

---

> ### Author Response · Authors · 2025-11-20
> **Response to Reviewer EDhE (4/n)**
>
> > **3. (Minor) In Section 5.3, the authors refer to an "ensemble" of evaluators. This term is usually reserved for a collection of different models or algorithms applied to the same task, rather than for multiple samples from a single algorithm.**
>
> Thanks for pointing it out! We will clarify in the paper that what we mean by ensemble is to run the same model multiple times with a non-zero sampling temperature, rather than running different models.
>
> ---
> ### Questions
>
> > **The authors describe the marking scheme generation methodology in Section 3.1 and later in Appendix A.3. However, the description lacks sufficient detail. Can the authors clarify: The model families they tested for rubric generation? what the prompts (with and without examples) entailed? How the annotators interacted with the initially generated rubrics? What was the average rubric quality rating in the final iteration? How was the best configuration selected, was it based solely on the average score?**
>
> We agree that our original submission lacked sufficient detail on the marking scheme generation process. In the revised paper, we have expanded this part in Section 2 and Appendix A.3, and clarified that marking scheme generation is part of our dataset construction rather than our core methodology. Below we summarize the requested details:
>
> * **Model families tested:** We compared o3 and Gemini 2.5 Pro, and selected the latter.
> * **Prompts (with and without examples):** These correspond to few-shot and zero-shot prompting, respectively. The marking scheme format follows [1]. Because no official rubrics from the target competitions are public, we used USEMO rubrics (https://web.evanchen.cc/usemo.html) as examples in the few-shot prompt. We included both prompts in Appendix A.10.
> * **Annotator interaction:** Annotators compared two marking schemes at a time (differing by model or prompt) using an interface shown in Figure 5. Two experts rated each scheme on a 0–3 scale and resolved disagreements greater than 1 point through discussion.
>    * *Stage 1 (model selection):* 18 problems (36 marking schemes). Zero-shot prompting was used. Annotators only differed more than 1 point in three cases and overall preferred Gemini 2.5 Pro. We revised the prompts based on annotators’ comments.
>    * *Stage 2 (prompt selection):* 36 problems. They both rated the zero-shot prompting slightly better. Feedback from stage 2 was used to further improve the prompt.
> * **Average rubric quality:** Of the 36 rubrics generated in Stage 2, 35 were rated 2 or 3 by both annotators. We didn’t grade rubrics for all problems.
> * **Configuration selection:** The final configuration was chosen based on annotator scores and qualitative feedback; while no setup produced perfect rubrics for all problems, we selected the overall best-performing one.
>
> Please find more details and annotator comments Appendix A.3 of the revised paper. We also provided the instruction document we gave to the annotators as a supplementary material.
>
> ---
> > **The authors have measured the bias of the graders with respect to the human grades. However, when breaking down by model, [1] and [2] show a clear positive bias toward a model's own solutions. Can the authors report similar metrics and discuss the implications?**
>
> This is a great point! Below we present generator vs. evaluator tables for two metrics: MAE and bias. MAE captures the deviation from expert scores, while bias measures the average signed error (i.e., systematic over- or under-scoring). Each row corresponds to an evaluator, and each column corresponds to a generator.
>
> **MAE:**
> | Evaluator/Generator |   o3  | Gemini 2.5 Pro | DeepSeek-R1 |
> |---------------------|------|---------------|------------|
> | o3                  | **1.120** | 0.778          | 0.993       |
> | Gemini 2.5 Pro      | 1.225 | **1.616**          | 1.183       |
> | DeepSeek-R1         | 1.181 | 1.432        | **1.504**       |
>
> **Bias:**
>
> | Evaluator/Generator |    o3  | Gemini 2.5 Pro | DeepSeek-R1 |
> |---------------------|-------|---------------|------------|
> | o3                  | -0.366 | **0.313**          | -0.070      |
> | Gemini 2.5 Pro      | -0.092 | **1.419**          | 0.451       |
> | DeepSeek-R1         | 0.226  | 0.959          | **0.989**       |
>
> The highest MAE and positive bias for each evaluator is bolded in the tables.
>
> Based on the results, each evaluator exhibits its highest MAE on outputs produced by the same model, indicating a tendency toward within-generator underperformance. In terms of “positive bias,” Gemini 2.5 Pro and DeepSeek-R1 tend to overscore their own responses, whereas the o3 evaluator does not exhibit this behavior. While the effect is modest in magnitude, it represents an important consideration for practical deployment and suggests that using diverse evaluator models or ensembles may help mitigate this systematic bias.

---

> ### Author Response · Authors · 2025-11-20
> **Response to Reviewer EDhE (5/5)**
>
> > **In what realistic setting can the authors' framework from Section 6 be applied (refer to comments in W2.3)?**
>
> Thanks for the question. As clarified in our W2.3 response, this section evaluates an evaluator’s ability to rank and select higher-quality proofs (not to propose a deployable RL reward model). Concretely, the same best-of-$n$ framework is realistic in the following settings:
>
> * **Rejection Sampling / Distillation.** Run the stronger (multi-pass) evaluator offline on large pools of generated proofs to filter or score them before training. This reduces human grading load and improves the quality of supervised/RL datasets.
> * **Training reward-model.** Use the full evaluator offline to label pairwise preferences or fine-grained scores, then train a cheaper reward/ranker that can be used online in best-of-n selection or lightweight RL.
> * **Human-in-the-loop triage.** In grading assistants or tutoring tools, use the evaluator to prioritize likely-correct or near-correct proofs for human review, while flagging low-quality ones for targeted feedback.
>
> ---
> > **When evaluating different provers, the authors have constrained themselves to using a score-based selection system. How does this compare to using a tournament-style approach?**
>
> Please refer to Figure 3 and Section 4.2 of our revised paper. The exhaustive tournament-style selection approach is highly effective and performs competitively compared to ProofGrader; however, it becomes computationally expensive when $n$ is large.
>
> ---
> > **The best-of-n evaluation was done on 29 selected problems. How were these problems selected and what was their difficulty distribution?**
>
> The 29 problems are from the 2025 contests, including the IMO, APMO, EGMO, USAMO, and USATST. Putnam was excluded as it had not yet been held at the time of submission. We did not select problems based on difficulty, so their difficulty distribution is similar to that of the full dataset except that they are more recent and non-contaminated.
>
> ---
> > **The authors claim in 3.2.3 that Staged Evaluation is "particularly effective for improving the performance of weaker backbone models." This is a very strong claim, given that this observation is only seen for the ***o4-mini*** model. Can the authors consider running additional experiments to support this statement or temper the claim to reflect the limited evidence?**
>
> This is a valid concern. We have added results for two additional weaker models (DeepSeek-R1-0528 and Qwen3-235B-A22B-Thinking) to address this point.
>
> | Model         | Design                | RMSE | MAE  | WTA(<=1) | Kendall-tau | Bias  |
> |--------------|-----------------------|-----:|-----:|---------:|------------:|------:|
> | DeepSeek-R1-0528 | Single pass (Ref+MS) | 1.735 | 1.357 | 66.4%     | **0.429**       | 0.732 |
> | DeepSeek-R1-0528 | Two Step              | **1.708** | **1.299** | **68.4%**     | 0.371       | **0.588** |
> | Qwen3-235B-A22B  | Single pass (Ref+MS) | 2.007 | 1.531 | 65.3%    | 0.423       | 0.724 |
> | Qwen3-235B-A22B  | Two Step              | **1.520** | **1.245** | **67.6%**    | **0.486**       | **0.333** |
>
> We have also revised our claim in Section 3.4 to the more precise statement that “staged pipelines may improve weaker models.” We believe the updated results sufficiently support this claim.
>
> ---
> > **For 2/3 models in 5.2.2, the *Strict* setting yields more accurate judgement than the Norm one. Do the authors have any qualitative explanations for this?**
>
> Yes. Qualitatively, we find that the difference reflects model strength rather than an inherent superiority of *Strict*. In our setting, truly correct but non-canonical solutions are relatively rare, while near-miss or superficially plausible but incorrect proofs are common. Under the *Norm* instructions, the evaluator is explicitly told to allow alternative approaches and flexibly map them to rubric checkpoints. For mid-tier models, this often leads to over-crediting: they treat near-miss solutions (“different but flawed”) as “different but valid” and assign scores that are too high. *Strict* instructions, in contrast, anchor these models more tightly to the marking scheme and penalize deviations, which reduces over-crediting and improves calibration for 2/3 models.
>
> ---
> [1] Evan Chen. Guidance for problem captains (Or: how to write an olympiad rubric). https://web.evanchen.cc/static/usemo/captain-guidance-usemo.pdf.

---

> ### Comment · Reviewer_EDhE · 2025-11-21
>
> I thank the authors for their concrete and thorough response. I specifically appreciate their effort on:
>
> - Providing the dataset and instructions for rubric creation for reproducibility.
> - Including further evaluation into the marking schema development.
> - Incorporating different experiments from my and other reviewers' recommendations.
>
> I appreciate the effort taken in such a short scope of time. I feel most of my concerns were addressed, and I have raised my score from 4 to 6. I am further delighted that the authors have already included parts of their response in the paper.
>
> I have 1 remaining concern regarding the practicality of the best-of-n experiment and a small comment regarding the novelty and scope of the dataset.
>
> - The authors propose that the best-of-n results highlight which models can be used in the following manners:
>     - **Rejection Sampling/Distillation and Training reward-model**: I agree that the results are practically valid in the proposed settings, and the verifiers can likely be highly effective. That said, the cost for running `ProofGrader` on even 1000 samples, if I am interpolating correctly will be ~$260, which can be quite expensive in non-proprietary model development.
>    - **Human-in-the-loop triage**: This is also a good idea, however, this particular implementation would require not just showing the results of the best solution, as in the best-of-n setting, but also how good the system is at ranking the solutions as well.
>
> - The dataset can in its current form only be used as a benchmark due to the small (though high-quality) number of samples. This is similar to [1], where the authors perform a similar LLM-as-a-judge task (though on the much easier binary classification task), but also provide a substantially larger dataset that could be used for training.
>
> Finally, I have one more question out of curiosity (which the authors shouldn't feel obliged to respond to):
>
> - Given that the new results in the rebuttal (Q2) show that models underperform as evaluators on their own solutions, would (and if so by how much) the performance improve if you use an ensemble of **different** evaluators that either (1) judge only other models' solutions (in), or if no knowledge of the solution source is assumed (2) a consensus/mean vote between `o3`, `Gemini-2.5-Pro`, and `DS-R1`.
>
> Depending on the discussions with the remaining reviewers, I will deliberate raising my score further throughout the discussion period.
>
> [1] Dekoninck et al. The open proof corpus: A large-scale study of llm-generated mathematical proofs. arXiv preprint arXiv:2506.21621, 2025.

---

> ### Author Response · Authors · 2025-11-23
> **Response to Reviewer EDhE (1/2)**
>
> We thank Reviewer EDhE for carefully reading our response, acknowledging our efforts, and re-evaluating our work. The new questions raised are thoughtful, and we provide our insights below. We are happy to discuss further with the reviewers.
>
> > **Rejection Sampling/Distillation and Training reward-model: the cost for running ProofGrader on even 1000 samples, if I am interpolating correctly will be ~$260, which can be quite expensive in non-proprietary model development.**
>
> We agree that the cost of running ProofGrader is non-trivial, and your estimate of roughly \$260 for 1000 evaluations in our current setup is correct. However, we believe this can be a manageable investment in the context of non-proprietary model development, particularly when viewed as a one-time cost for data generation rather than a recurring inference cost. First, training competitive open-source models with RL typically requires many GPU-days, which corresponds to costs that are orders of magnitude higher than a few thousand ProofGrader calls. For example, renting a single H100 node for one day can already exceed \\$500. For distillation pipelines where training is less expensive, the single-pass variant of ProofGrader remains a strong and cheaper alternative. Second, obtaining human labels of comparable quality is significantly more expensive. Even at \\$30/h for undergraduate graders (and much higher rates for IMO medalists or expert coaches), manually grading 1000 competition-style proofs would far exceed \\$260.
>
> We also note that the cost of running strong reasoning models is decreasing rapidly. For example, the o-series models have seen dramatic price reductions: o3’s API pricing dropped by nearly 80\% to \\$2 per million input tokens and \\$8 per million output tokens *within two months of release* [2]. In less than a year, o3-pro became roughly 8× cheaper than o1-pro. This trend suggests inference costs for high-quality verifiers may drop *far faster than GPU prices or human annotation costs*, which tend to remain relatively stable.
>
> > **Human-in-the-loop triage: this particular implementation would require not just showing the results of the best solution, as in the best-of-n setting, but also how good the system is at ranking the solutions as well.**
>
> This is a great point, and we agree that for human-in-the-loop triage it is important to evaluate not only whether the system can pick the best solution, but also how well it ranks candidate solutions overall. We have run such analyses on the best-of-n datasets for ProofGrader. Specifically, we measure:
> - **Pairwise comparison accuracy.** For each pair of responses to the same problem, we check whether ProofGrader’s preference matches the human preference, skipping ties. On 1912 pairs, the macro-averaged accuracy is 0.813 (std = 0.142).
> - **Spearman rank correlation.** We compare the ranking induced by ProofGrader’s scores to the ranking from human scores. The macro-averaged Spearman ρ is 0.805 (std = 0.157).
>
> These results indicate that ProofGrader not only selects strong candidates in the best-of-n setting, but also produces rankings that are largely consistent with human preferences, which supports its suitability for human-in-the-loop triage.
>
> > **The dataset can in its current form only be used as a benchmark due to the small (though high-quality) number of samples. This is similar to [1], where the authors perform a similar LLM-as-a-judge task (though on the much easier binary classification task), but also provide a substantially larger dataset that could be used for training.**
>
> We thank the reviewer for pointing out [1], which is closely related and concurrent work. We have discussed [1] in multiple sections of the paper, and we will further clarify the relationship. While [1] focuses on building a large-scale dataset for training and benchmarking proof judges with binary correctness labels, our paper focuses on **designing and analyzing fine-grained evaluators**, with ProofBench serving primarily as the **testbed** that makes this study possible. In other words, the benchmark is an enabling component rather than the main end goal of our work.
>
> We agree that, in its current form, ProofBench is better viewed as a high-quality benchmark than as a large training corpus. While smaller in scale than [1], constructing a competition-style, fine-grained dataset with 0–7 scores aligned to problem-specific marking schemes and expert consensus is itself non-trivial and, we believe, valuable to the community, both for our evaluator study and for future work on proof evaluation. This reflects how we intend to position ProofBench in the paper. Importantly, our contributions go beyond simply converting binary labels to a 0–7 scale. We empirically investigate how evaluator design affects the reliability of proof evaluation and, through best-of-n experiments, show that *fine-grained scoring is preferable* for reliably selecting higher-quality proofs.

---

> > ### Author Response · Authors · 2025-11-23
> > **Response to Reviewer EDhE (2/2)**
> >
> > > **Given that the new results in the rebuttal (Q2) show that models underperform as evaluators on their own solutions, would (and if so by how much) the performance improve if you use an ensemble of different evaluators that either (1) judge only other models' solutions (in), or if no knowledge of the solution source is assumed (2) a consensus/mean vote between o3, Gemini-2.5-Pro, and DS-R1.**
> >
> > This is a great suggestion. We conducted exploratory experiments to evaluate exactly these two ensemble settings. As noted in our response to Q2, **o3 remains the strongest single evaluator across all generators**, even though its MAE is highest on its own solutions. To more directly answer the reviewer’s question, we expanded the generator–evaluator MAE matrix to include simple mean-score ensembles of pairs and triplets of evaluators.
> >
> > | Evaluator | o3 | Gemini | R1 |
> > |-----------|-------|--------|-------|
> > | o3 | 1.120 | 0.778 | **0.993** |
> > | Gemini | 1.225 | 1.616 | 1.183 |
> > | R1 | 1.181 | 1.432 | 1.504 |
> > | o3+Gemini | 1.125 | 1.236 | 1.180 |
> > | o3+R1 | 1.201 | 1.301 | 1.225 |
> > | Gemini+R1 | 1.282 | 1.282 | 1.261 |
> > | o3+Gemini+R1 | 1.168 | 1.232 | 1.198 |
> > | ProofGrader (o3-ensemble) | **1.099** | **0.768** | 1.028 |
> > (Note: lowest MAE is bolded for each generator.)
> >
> > The results reveal two patterns:
> > - **Naive ensembling does not generally improve accuracy.** Mean aggregation of different evaluators (e.g., o3+Gemini or Gemini+R1) often underperforms the strongest individual evaluator. In the reviewer’s setting (1), Gemini/R1/Gemini+R1 does not outperform o3 on o3-generated solutions. In setting (2), aggregating three evaluators fails to surpass the single best model.
> > - **Performance is still dominated by o3.** Across all generators, the single-pass o3 evaluator or the o3-based ProofGrader achieves the best MAE. Adding weaker evaluators tends to dilute o3’s strength rather than enhance it.
> >
> > One hypothesis is that ensembles help only when the underlying evaluators are comparably strong and have complementary error profiles. Given the current gap between o3 and other models, naive score averaging does not yield gains. More sophisticated approaches, such as multi-agent debate, may offer gains beyond simple averaging, but this is outside the scope of this work.
> >
> > ---
> > [1] Dekoninck et al. The open proof corpus: A large-scale study of llm-generated mathematical proofs. arXiv preprint arXiv:2506.21621, 2025.
> >
> > [2] X post by OpenAI Developers. https://x.com/OpenAIDevs/status/1932532777565446348. Jun 2025.

---

> > > ### Comment · Reviewer_EDhE · 2025-11-26
> > >
> > > I thank the reviewers for their follow-up! I believe my concerns have been sufficiently addressed. I will go through the discussions with other reviewers by the end of the discussion period, and update my summary and my evaluation accordingly.

---

### Official Review · Reviewer_zjS9 · 2025-10-27

**Soundness:** 4
**Presentation:** 2
**Contribution:** 4
**Rating:** 6
**Confidence:** 4

**Summary:**

This paper tackles the critical bottleneck of evaluating natural language math proofs from LLMs with PROOFBENCH, a new expert-annotated dataset of LLM-generated proofs graded on a 0–7 scale using detailed marking schemes. Based on a systematic study of evaluator design, they developed PROOFGRADER, an evaluator that achieves high alignment with human experts. Moreover, they show in a Best-of-N task where PROOFGRADER, as a reward model, closes over 90% of the performance gap between a naive binary evaluator and a human oracle.

**Strengths:**

1. The paper tackles the crucial and timely problem of scalable, reliable proof evaluation.
2. PROOFBENCH represents a substantial and meticulous human annotation effort.
3. The BoN experiments demonstrate the effectiveness of the methods.

**Weaknesses:**

1. The paper relies heavily on aggregate metrics (RMSE, etc.). It would be better to include some case studies and error analyses.
2. The paper mentions MathArena but doesn't adequately compare PROOFBENCH to MathArena's manual annotation efforts.
3. The backbone models are mainly proprietary models. The analysis would be more comprehensive if it evaluated some open-source models (e.g., Llama and Qwen series) to look into their capabilities as evaluators.

**Questions:**

1. Does the evaluator show a "self-enhancement bias"? For instance, does the O3-backbone evaluator systematically over-score proofs generated by O3? A heatmap of bias (Generator vs. Evaluator) would be insightful.
2. The paper shows that strong models (like O3) are strong evaluators. What about the other dynamics? How well do weak models evaluate strong models, and vice versa? Understanding this is important for understanding the robustness of "LLM-as-a-judge" and its potential for "weak to strong generalization".
3. Is there a risk that generator-LLMs could hack the evaluator to get a high score?

---

> ### Author Response · Authors · 2025-11-20
> **Response to Reviewer zjS9 (1/n)**
>
> We thank Reviewer zjS9 for the valuable feedback! Below we address the questions and concerns raised by the reviewer.
>
> > **W1:The paper relies heavily on aggregate metrics (RMSE, etc.). It would be better to include some case studies and error analyses.**
>
> This is a valid concern. Please refer to our global response question #2.
>
> > **W2: The paper mentions MathArena but doesn't adequately compare ProofBench to MathArena's manual annotation efforts.**
>
> We agree that MathArena, which also targets proof-based evaluation, is a closely related effort. Below we add a concise comparison on our overlapping contests, IMO 2025 and USAMO 2025, using the three shared models (o3, Gemini 2.5 Pro, DeepSeek-R1-0528). We report normalized contest scores as percentages of the maximum possible points:
>
> **IMO 2025**
>
> |  | o3 | Gemini 2.5 Pro | DeepSeek-R1-0528 |
> |---|---|---|---|
> | **ProofBench** | 14.29% | 21.42% | 11.90% |
> | **MathArena** | 16.67% | 31.55% | 6.85% |
>
> **USAMO 2025**
>
> |  | o3 | Gemini 2.5 Pro | DeepSeek-R1-0528 |
> |---|---|---|---|
> | **ProofBench** | 33.33% | 23.80% | 9.52% |
> | **MathArena** | N/A | 24.40% | 30.06% |
>
> On IMO 2025, both frameworks agree on the ranking (Gemini > o3 > DeepSeek), though absolute percentages differ. On USAMO 2025, rankings diverge more, especially for DeepSeek-R1-0528. A key difference is that MathArena averages over 4 samples per problem, while ProofBench uses a single sample, which tends to penalize DeepSeek’s frequent near-miss attempts more strongly.
>
> Overall, MathArena focuses on fresh problems from recent contests with multi-sample evaluation, while ProofBench offers broader coverage across multiple contests and years with fine-grained, rubric-based scores. We view the two benchmarks as complementary.
>
> > **W3: The backbone models are mainly proprietary models. The analysis would be more comprehensive if it evaluated some open-source models (e.g., Llama and Qwen series) to look into their capabilities as evaluators.**
>
> We would like to clarify that our original submission already included DeepSeek-R1-0528, one of the strongest open-source reasoning models available at the time of submission.
>
> To address your feedback more comprehensively, we have conducted additional experiments with leading open-source models at different capability levels: **Qwen3-235B-A22B (Thinking)** and **Llama-3.1-70B-Instruct**.
>
> | Model | Context | MAE | RMSE | Bias | WTA(<=1) |
> |---|---|---|---|---|---|
> | Llama-3.1-70B-Instruct | Ref+MS | 3.212 | 3.679 | 3.156 | 31.0% |
> | Llama-3.1-70B-Instruct | MS | 3.189 | 3.385 | 3.150 | 27.9% |
> | Llama-3.1-70B-Instruct | Ref | 3.194 | 3.661 | 3.096 | 31.2% |
> | Llama-3.1-70B-Instruct | None | 3.945 | 4.404 | 3.898 | 26.1% |
> | Qwen3-235B-A22B | Ref+MS | 1.531 | 2.007 | 0.724 | 65.3% |
> | Qwen3-235B-A22B | MS | 1.351 | 1.727 | 0.041 | 66.8% |
> | Qwen3-235B-A22B | Ref | 1.860 | 2.068 | 1.425 | 45.3% |
> | Qwen3-235B-A22B | None | 2.631 | 3.070 | 2.359 | 40.6% |
>
> Unfortunately, these models demonstrate much weaker performance as evaluators compared to frontier proprietary models. Given that these represent some of the strongest available open-source models in their respective families, evaluating smaller variants (e.g., 7B or 14B models) would likely yield even weaker results. These findings suggest that current open-source models significantly lag behind proprietary models in fine-grained mathematical proof evaluation. The results are included in Table 13 of section A.7 in the revised paper.

---

> > ### Author Response · Authors · 2025-11-20
> > **Response to Reviewer zjS9 (2/2)**
> >
> > > **Q1: Does the evaluator show a "self-enhancement bias"? For instance, does the O3-backbone evaluator systematically over-score proofs generated by O3? A heatmap of bias (Generator vs. Evaluator) would be insightful.**
> >
> > This is a great observation! Below we present generator vs. evaluator tables for two metrics: MAE and bias. MAE captures the deviation from expert scores, while bias measures the average signed error (i.e., systematic over- or under-scoring). Each row corresponds to an evaluator, and each column corresponds to a generator.
> >
> > **MAE**
> >
> > | Evaluator/Generator |    o3 | Gemini 2.5 Pro | DeepSeek-R1 |
> > | ------------------- | ---- | ------------- | ---------- |
> > | o3                  | **1.120** |          0.778 |       0.993 |
> > | Gemini 2.5 Pro      | 1.225 |          **1.616** |       1.183 |
> > | DeepSeek-R1         | 1.181 |          1.432 |       **1.504** |
> >
> > **Bias**
> >
> > | Evaluator/Generator |     o3 | Gemini 2.5 Pro | DeepSeek-R1 |
> > | ------------------- | ----- | ------------- | ---------- |
> > | o3                  | -0.366 |          **0.313** |      -0.070 |
> > | Gemini 2.5 Pro      | -0.092 |          **1.419** |       0.451 |
> > | DeepSeek-R1         |  0.226 |          0.959 |       **0.989** |
> >
> > The highest MAE and positive bias for each evaluator is bolded in the tables.
> >
> > Based on the results, each evaluator exhibits its highest MAE on outputs produced by the same model, indicating a tendency toward within-generator underperformance. In terms of “self-enhancement bias,” Gemini 2.5 Pro and DeepSeek-R1 tend to overscore their own responses, whereas the o3 evaluator does not exhibit this behavior. Notably, the o3-based evaluator remains the strongest across all generators.
> >
> > We have included these results in section 3.3 and Table 3 in the revised paper.
> >
> > > **Q2: The paper shows that strong models (like O3) are strong evaluators. What about the other dynamics? How well do weak models evaluate strong models, and vice versa? Understanding this is important for understanding the robustness of "LLM-as-a-judge" and its potential for "weak to strong generalization".**
> >
> > This is a great question about cross-capability dynamics. We have conducted additional experiments to examine both directions: strong evaluators on weak generators, and weak evaluators on strong generators.
> >
> > **Strong evaluators on weak generators:** We leverage pilot data from two weaker models, Qwen3-4B and DeepSeek-R1-0528-Qwen3-8B, on 36 problems (full annotation on additional models requires substantial effort). Both of them are among the strongest available reasoning models of the same scale. Results are shown below.
> >
> > | Generator                 | Average score (out of 7) | 0-point solutions |
> > |--------------------------|--------------------------|------------------|
> > | Qwen3-4B                 | 0.92                      | 70%               |
> > | DeepSeek-R1-0528-Qwen3-8B| 0.78                      | 72%               |
> >
> > | Evaluator     | MAE   | RMSE  | WTA(<=1) |
> > |---------------|------|------|---------|
> > | **ProofGrader**   | **0.431** | **0.521** | 93.1%    |
> > | o3 (Ref+MS)   | 0.486 | 0.559 | **95.8%**    |
> > | o3 (None)     | 1.764 | 1.892 | 59.7%    |
> >
> > We find that the strong evaluators grade weaker models more accurately because these models mostly produce incorrect (0-point) proofs, and they align more closely with human experts on low-quality proofs than on higher-quality ones. Interestingly, strong evaluators also achieve their highest accuracy when grading clearly incorrect solutions (0 points) from strong generators, suggesting that evaluation difficulty increases primarily for partial-credit and near-complete solutions.
> >
> > **Weak evaluators on strong generators:** As mentioned in our response to your earlier question, we evaluated weaker open-source models (Qwen3-235B-A22B and Llama-3.1-70B-Instruct) as evaluators on solutions from strong generators. These weak evaluators perform poorly. This indicates significant challenges for "weak-to-strong generalization" in mathematical proof evaluation.
> >
> > Together, these findings suggest an asymmetry: strong evaluators can reliably grade weak generators, but weak evaluators cannot reliably grade strong generators.
> >
> > > **Q3: Is there a risk that generator-LLMs could hack the evaluator to get a high score?**
> >
> > In our current setup, generators are neither trained nor tuned on evaluator feedback: they are pretrained/finetuned independently, and the evaluator is used only at test time. This substantially limits opportunities for explicit reward hacking. Moreover, our evaluators grade for mathematical correctness and proof completeness, which makes them less sensitive to stylistic artifacts or superficial cues. While a generator might try to target anticipated “award points,” our checkpoints correspond to non-trivial intermediate results, so reverse-engineering them is roughly as hard as making genuine progress on the proof.

---

> ### Comment · Reviewer_zjS9 · 2025-11-21
>
> Thank you for your reply. I have raised my score from 6 to 8. I have one more small suggestion: it would be great to improve the aesthetics of Figure 2. Currently, the text is quite small and thin, and the colors are not very appealing.

---

> > ### Author Response · Authors · 2025-11-22
> > **Response to Reviewer zjS9**
> >
> > Thank you again for taking the time to read our response and re-evaluate our work! We appreciate your suggestion regarding Figure 2; we’ve uploaded a revised version that improves its clarity, readability, and overall presentation.

---

### Official Review · Reviewer_LGPu · 2025-10-30

**Soundness:** 3
**Presentation:** 3
**Contribution:** 3
**Rating:** 6
**Confidence:** 4

**Summary:**

This paper addresses the challenge of reliably evaluating natural language mathematical proofs generated by large language models. The authors propose a systematic methodology for developing automated evaluators that assign fine-grained scores on a 0-7 scale, rather than binary correct/incorrect judgments. They introduce PROOFBENCH, an expert-annotated dataset containing 393 LLM-generated solutions to 131 competition math problems, along with problem-specific marking schemes. Through systematic experimentation, they develop PROOFGRADER, an ensemble-based evaluator that achieves strong alignment with expert judgments (RMSE of 1.093) and demonstrates practical utility as a reward model for best-of-n selection tasks.

**Strengths:**

1 The work tackles an important gap in mathematical proof evaluation with a systematic approach. While building on existing LLM-as-a-judge paradigms, the application to fine-grained mathematical proof evaluation with problem-specific marking schemes is novel.
2. The experimental design is thorough and methodical. The expert annotation process is well-designed with appropriate quality controls. The systematic ablation studies provide clear insights about what factors matter for evaluator performance.
3. The paper is well-organized and clearly written. The motivation is compelling, methodology is systematic, and results are presented comprehensively with good visualizations.
4. Addresses a real bottleneck in mathematical reasoning research. PROOFBENCH provides a valuable resource for the community, and the insights about evaluator design will inform future work. The demonstration of practical utility in best-of-n selection shows real-world applicability.

**Weaknesses:**

1. The dataset contains only 393 solutions across 131 problems from competition mathematics. This is relatively small for drawing broad conclusions about evaluator design, and competition problems may not represent the full spectrum of mathematical reasoning tasks.
2, The approach relies heavily on LLM-generated marking schemes, which could introduce systematic biases or limitations. The quality of these schemes fundamentally constrains the evaluation quality, but this dependency is not thoroughly analyzed.
3. All experiments use the same expert annotators and marking scheme generation process. It's unclear how well the findings generalize to different mathematical domains, difficulty levels, or evaluation standards.
4. The paper doesn't compare against other potential evaluation approaches beyond varying LLM configurations. For instance, how does this approach compare to simpler heuristic methods or other structured evaluation frameworks?

**Questions:**

1. How sensitive are the results to the quality of the automatically generated marking schemes? Have you conducted experiments with human-written marking schemes for comparison?
2. How well do these evaluator design principles transfer to other mathematical domains beyond competition problems (e.g., research-level proofs, educational contexts)?
3. Can you provide more detailed analysis of when and why the evaluator fails? What types of mathematical reasoning or proof structures are most challenging?
4. What are the computational costs of your best evaluator compared to simpler alternatives? How does this scale with problem complexity?
5.What is the inter-annotator agreement between your experts, and how does this compare to the agreement between experts and your automated evaluator?

---

> ### Author Response · Authors · 2025-11-20
> **Response to Reviewer LGPu (1/n)**
>
> We thank Reviewer LGPu for the valuable feedback! We address the questions and concerns as follows.
>
> > **W1: The dataset contains only 393 solutions across 131 problems from competition mathematics. This is relatively small for drawing broad conclusions about evaluator design, and competition problems may not represent the full spectrum of mathematical reasoning tasks.**
>
> Please see the global response for updates to our dataset: we have expanded ProofBench to 145 problems with 435 solutions, covering all problems from six major competitions (USAMO, IMO, etc.) across four years (2022-2025).
>
> While the absolute data size is modest, we believe this scale is close to the maximum we can get, given the inherent scarcity of human competitions. Our data  already covers all major competitions and includes a diverse range of domains (e.g., algebra, geometry, combinatorics), difficulty levels (e.g., EGMO and APMO being easier, IMO and USATST being harder), and even educational stages (e.g., Putnam at the college level). Note that prior efforts typically covered a single competition, or even a single year [3,4].
>
> > **W2: The approach relies heavily on LLM-generated marking schemes, which could introduce systematic biases or limitations. The quality of these schemes fundamentally constrains the evaluation quality, but this dependency is not thoroughly analyzed.**
>
> We appreciate this important concern and have taken several steps to ensure marking scheme quality and to mitigate potential biases.
>
> First, we developed the marking scheme generator through a rigorous evaluation on 36 problems. We compared multiple backbone models (o3 vs. Gemini 2.5 Pro) and prompt styles (zero-shot vs. few-shot), with initial prompt templates informed by expert-written competition grading guidelines [1,2]. Two experts rated each marking scheme variant on a 0–3 scale, showed high agreement in their preferences, and provided iterative feedback that guided prompt refinements. Additional details are provided in Section A.3 of the revised paper.
>
> Second, annotators were explicitly instructed to treat generated marking schemes as flexible references, not rigid scoring rules. When point allocations seemed unreasonable or when a solution followed an approach not covered in the scheme, annotators were expected to exercise independent expert judgment. According to annotator reports, the generated marking schemes were reasonable and helpful in over 85% of cases.
>
> We acknowledge that using expert-written, official competition marking schemes could have been better. However, official competitions do not publicly release their grading schemes, and manually creating problem-specific marking schemes requires specialized training that was not accessible for our study. We view this as an important direction for future work to develop more reliable and scalable marking scheme generation methods. Within current practical constraints, our approach with quality controls represents a reasonable and transparent methodology.
>
> > **W3: All experiments use the same expert annotators and marking scheme generation process. It's unclear how well the findings generalize to different mathematical domains, difficulty levels, or evaluation standards.**
>
> We thank the reviewer for this valuable point. Firstly, math competitions in ProofBench already cover a diverse range of domains (e.g., algebra, geometry, combinatorics), difficulty levels (e.g., EGMO and APMO being easier, IMO and USATST being harder), and even educational stages (e.g., Putnam at the college level).
>
> Our focus is specifically on competition-style proofs, which are relatively short and self-contained. Within this setting, we show that fine-grained scoring and rubric-based evaluation consistently improve evaluator reliability across multiple contests and years. We view these as plausible general principles for proof evaluation, but we do not claim they transfer unchanged to research-level or highly open-ended proofs, which raise additional challenges (e.g., much longer context). In such settings, the expert pool and rubric-generation instructions would need to be adapted, but the overall study pipeline remains applicable.
> We will clarify this limitation in the revised paper and explicitly highlight adapting our framework to educational and research settings as an important direction for future work.

---

> > ### Author Response · Authors · 2025-11-20
> > **Response to Reviewer LGPu (2/2)**
> >
> > > **W4: The paper doesn't compare against other potential evaluation approaches beyond varying LLM configurations. For instance, how does this approach compare to simpler heuristic methods or other structured evaluation frameworks?**
> >
> > Since we are trying to evaluate free form, natural language solutions to proof-based math problems, simple heuristic methods (answer exact match, length features, key word counting, etc) are not viable substitutes for grading: many problems do not have easily checkable final answers; those methods cannot reliably distinguish partially correct reasoning, or subtle logical errors, which are exactly what competition-style marking schemes are designed to capture. We mentioned this in the beginning of section 1.
> >
> > Other structured frameworks, such as formal proof checkers, typically require the solutions to be written in a specific formal language and are not applicable to our data, where solutions are in natural language. We will clarify this scope in the revision, while highlighting integration with other heuristic or formal methods as an interesting but orthogonal direction for future work.
> >
> > Therefore, our goal in this work is to study design choices with LLM-based evaluators for natural language competition proofs, rather than claiming superiority over all other evaluation paradigms. If you have some specific alternative evaluation methods in mind, could you please specify them? We will be happy to run for comparisons.
> >
> > > **Q1: How sensitive are the results to the quality of the automatically generated marking schemes? Have you conducted experiments with human-written marking schemes for comparison?**
> >
> > Thank you for the question. For the competitions included in ProofBench, official human-written marking schemes are not publicly released, so we cannot run a direct controlled comparison with these schemes; producing new human-written schemes would require trained contest graders and was beyond our resources. Instead, we validated our marking-scheme generator in a closely matched setting: for USEMO (a competition with similar style and standards), experts compared model-generated schemes against human-written ones, rating each on a 0-3 quality scale and iteratively providing feedback. In the end, 35 of 36 generated marking schemes received scores of 2 or 3, indicating they are largely comparable to human-written ones. Supported by this strong validation process, we believe that our reported results are robust and reflect realistic deployment performance. We detailed the validation process in section A.3 of the revised paper.
> >
> > > **Q2: How well do these evaluator design principles transfer to other mathematical domains beyond competition problems (e.g., research-level proofs, educational contexts)?**
> > Please first refer to our response to W3, which discusses domain coverage and generalization within competition mathematics.
> >
> > At a high level, we believe the overall principles (constructing datasets, generating marking schemes, and designing LLM-based evaluators) can transfer to other mathematical domains, including educational settings. For example, college-level homework and exam grading similarly relies on structured rubrics, reference solutions, and fine-grained point allocations. In such contexts, the same pipeline could be applied, though the specific rubric instructions, difficulty calibrations, and domain conventions would naturally differ.
> >
> > However, collecting high-quality data outside competition settings requires substantial expert effort and falls beyond the scope of this paper. We therefore have not validated these extensions empirically. We will clarify this limitation in the revised version and highlight broader-domain adaptation as a valuable direction for future work.
> >
> > > **Q3: Can you provide more detailed analyses of when and why the evaluator fails? What types of mathematical reasoning or proof structures are most challenging?”**
> >
> > Please refer to our global response question #2.
> >
> > >  **Q4: What are the computational costs of your best evaluator compared to simpler alternatives? How does this scale with problem complexity?**
> >
> > Please refer to our global response question #3.
> >
> > > **Q5: What is the inter-annotator agreement between your experts, and how does this compare to the agreement between experts and your automated evaluator?**
> >
> > Please refer to our global response question #1.
> >
> > ---
> >
> > [1] Evan Chen. Guidance for problem captains (Or: how to write an olympiad rubric). https://web.evanchen.cc/static/usemo/captain-guidance-usemo.pdf.
> >
> > [2] USEMO resources. https://web.evanchen.cc/usemo.html.
> >
> > [3] Petro et al. Proof or Bluff? Evaluating LLMs on 2025 USA Math Olympiad. https://arxiv.org/abs/2503.21934.
> >
> > [4] Balunović et al. MathArena: Evaluating LLMs on Uncontaminated Math Competitions. https://arxiv.org/abs/2505.23281.

---

> ### Author Response · Authors · 2025-11-26
> **Official Comment by Authors**
>
> Dear Reviewer LGPu,
>
> With the December 2 rebuttal deadline approaching, we wanted to reach out to see whether you have any follow-up questions or clarifications regarding our submission. We would be glad to address any remaining concerns and provide further explanation where helpful. If possible, we would appreciate receiving any comments at your earliest convenience.
>
> Thank you again for your time and thoughtful reviews.
>
> Best regards,
>
> The Authors

---

### Official Review · Reviewer_HNs6 · 2025-11-01

**Soundness:** 2
**Presentation:** 2
**Contribution:** 3
**Rating:** 4
**Confidence:** 3

**Summary:**

This paper introduces ProofBench, a dataset of 131 competition math problems with 393 expert-graded LLM solutions, and systematically studies design choices for automated proof evaluators. The authors propose ProofGrader, which achieves RMSE of 1.093 against expert scores and demonstrates practical utility in best-of-n selection tasks.

**Strengths:**

S1: The paper tackles an important problem for mathematical reasoning research. Reliable proof evaluation is a critical bottleneck for training and assessing LLMs on mathematical reasoning tasks, and the lack of scalable alternatives to human grading or formal verification makes this work timely and valuable.

S2: The systematic methodology using problem-specific marking schemes as an anchor is well-motivated. The two-stage annotation process (generating marking schemes, then grading with them as guidance) provides a principled way to maintain consistency while allowing flexibility for alternative solution approaches.

S3: The experimental design is comprehensive and well-structured. The ablation studies clearly isolate the impact of different design choices (backbone model, context components, instructions, workflows), providing actionable insights about what matters for evaluator performance.

S4: The best-of-n experiment (Section 6) effectively demonstrates practical utility beyond correlation metrics. Showing that ProofGrader closes 90% of the gap between a naive binary evaluator and the human oracle provides evidence of real-world value for downstream applications like RL training.

S5: The decision to use fine-grained (0-7) rather than binary scoring is validated empirically. Figure 2 clearly shows that binary evaluators fail to distinguish among correct solutions, while fine-grained scoring enables effective ranking.

**Weaknesses:**

W1: ProofBench is only shown to show predictive power (via MSE on the benchmark) over whether an evaluator is good at grading outputs of three specific models (o3, Gemini 2.5 Pro, DeepSeek-R1), all of which are precisely the models whose proofs are used as the annotated examples for computing the MSE. There is no evaluation of how well ProofGrader generalizes to solutions from weaker models, different model families, or human-written proofs, which limits our understanding of its robustness. The paper should have tested whether evaluators with low MSE on ProofBench have high agreement with human annotators on grading proofs generated by models that are not one of those three models, in order to demonstrate general utility.

W2: Inter-annotator reliability is insufficiently reported. While the paper mentions that two experts underwent calibration and double-scored 20% of items, the actual agreement metrics (e.g., correlation, exact agreement rate, within-1 agreement) are not provided. Since the significance of the contributions hinges on the correctness of the human annotations, this needs to be emphasized in the main text.

W3: Error analysis is absent. The paper does not systematically examine where ProofGrader fails or succeeds, what types of errors it makes (over-crediting vs. under-crediting), or which problem types are most challenging. Understanding failure modes would be valuable for future work and practical deployment.

W4: Computational costs and efficiency are not discussed. For practical deployment, especially in RL training scenarios where evaluators may need to score thousands of candidate solutions, the cost (in terms of API calls, latency, and financial expense) compared to simpler baselines would be relevant information.

**Questions:**

Q1. Since the best-of-n experiments are supposed to establish the predictive generalization power of ProofBench for use as a reward model, shouldn’t the ground truth annotations for the proofs used in the experiment be annotated by different people than those used to annotate the examples in ProofBench? A difficulty in designing a benchmark for evaluators is that the bias in the annotations themselves need to be accounted for, i.e, the paper needs to demonstrate that the possibly imperfect annotator preferences used in ProofBench are sufficient for generalization to general aesthetic preferences in the math community (regarding competition style proofs).

Q2. Similarly to Q1, in the best-of-n experiments, shouldn't the set of model generations being evaluated by the candidate evaluators be generated by models that are different to the three specific models used to generate the outputs in the benchmark? Could the authors provide justification why the current methodology is sufficient to demonstrate the general utility of ProofBench for selecting evaluators meant to evaluate generations from other models? Currently, it seems like ProofBench is useful for selecting evaluators for proofs generated by o3, Gemini, and DeepSeek-R1, but no indication of general ability.

Q3. This approach seems very committal to a specific frozen mark scheme. Since the labor intensive annotation process is predicated on a particular MS, what happens if the mark scheme needs to be changed? Do we need to re-annotate each time? Could the authors provide justification why using a particular frozen mark scheme for the annotation process is sufficiently general to avoid re-annotation?

---

> ### Author Response · Authors · 2025-11-20
> **Response to Reviewer HNs6 (1/n)**
>
> We thank Reviewer HNs6 for recognizing the strengths and contributions of our paper and for providing such valuable and constructive feedback! Below, we address each weakness and question in the order they appear in your review.
>
> > **W1: ProofBench is only shown to show predictive power (via MSE on the benchmark) over whether an evaluator is good at grading outputs of three specific models (o3, Gemini 2.5 Pro, DeepSeek-R1), all of which are precisely the models whose proofs are used as the annotated examples for computing the MSE. There is no evaluation of how well ProofGrader generalizes to solutions from weaker models, different model families, or human-written proofs, which limits our understanding of its robustness. The paper should have tested whether evaluators with low MSE on ProofBench have high agreement with human annotators on grading proofs generated by models that are not one of those three models, in order to demonstrate general utility.**
>
> This is an important concern about generalization. Please note that ProofBench already includes three generators from different model families, spanning both proprietary and open-source models. To further support our claim, we also provide additional results on weaker open-source models below.
>
> **Evaluation on weaker, open-source models:** We collected expert gradings on solutions from two weaker models (Qwen3-4B and DeepSeek-R1-0528-Qwen3-8B) on a selected set of 36 problems from 2024 and 2025 competitions. Both models are among the best math reasoning models of the same scale. Here we provide the results on the generators and selected evaluators.
>
> | Generator | Average score (out of 7) | 0-point solutions |
> |---|---|---|
> | Qwen3-4B | 0.92 | 70% |
> | DeepSeek-R1-0528-Qwen3-8B | 0.78 | 72% |
>
> | Evaluator | MAE | RMSE | WTA(<=1) |
> |---|---|---|---|
> | **o3 ensemble (Ref+MS) – ProofGrader** | **0.431** | **0.521** | 93.1% |
> | o3 (Ref+MS) | 0.486 | 0.559 | **95.8%** |
> | o3 (None) | 1.764 | 1.892 | 59.7% |
> | DeepSeek-R1-0528 (Ref+MS) | 0.750 | 0.929 | 81.9% |
> | DeepSeek-R1-0528 (None) | 3.653 | 3.823 | 11.1% |
> | gpt-4o (Ref+MS) | 2.458 | 2.669 | 26.4% |
> | gpt-4o (None) | 3.931 | 4.056 | 5.6% |
>
> Proofgrader’s error rates are actually significantly lower here than those reported in the original paper. This is because weaker models mostly produce incorrect (0-point) proofs, making grading much easier. Since grading weaker models is generally easier, we chose to use the strongest available model as generators in ProofBench. This allows the benchmark to better challenge evaluators by requiring them to distinguish between partially correct and nearly-complete solutions, and to detect subtle errors in reasoning steps. We included this analysis in section A.6 of the revised paper.
>
> **Regarding generalization to human proofs:** While this is an interesting direction, it falls outside the scope of our work. Our goal is to develop reliable proof evaluation for assessing *LLMs*, as mentioned in Section 1. Furthermore, collecting human-written proofs across different levels of expertise is very challenging, since they’re mostly not publicly available. Extending our work to evaluate human solutions, e.g., for educational purposes, would be a valuable future direction.
>
> > **W2: Inter-annotator reliability is insufficiently reported. While the paper mentions that two experts underwent calibration and double-scored 20% of items, the actual agreement metrics (e.g., correlation, exact agreement rate, within-1 agreement) are not provided. Since the significance of the contributions hinges on the correctness of the human annotations, this needs to be emphasized in the main text.**
>
> Please refer to our global response (especially question #1): We have updated ProofBench so that 41% of the proofs are double-graded. The within-1-point agreement between experts is 87.5%, followed by discussion to reach final consensus on all scores.
>
> > **W3: Error analysis is absent. The paper does not systematically examine where ProofGrader fails or succeeds, what types of errors it makes (over-crediting vs. under-crediting), or which problem types are most challenging. Understanding failure modes would be valuable for future work and practical deployment.**
>
> Please refer to our global response question #2.

---

> > ### Author Response · Authors · 2025-11-20
> > **Response to Reviewer HNs6 (2/n)**
> >
> > > **W4: Computational costs and efficiency are not discussed. For practical deployment, especially in RL training scenarios where evaluators may need to score thousands of candidate solutions, the cost (in terms of API calls, latency, and financial expense) compared to simpler baselines would be relevant information.**
> >
> > This is a valid concern for practical deployment. For a detailed cost analysis, please see our global response to question #3.
> >
> > We agree with the reviewer that cost may become a concern when incorporating our method into RL training. However, in RL, rollout generation typically dominates training time, as rollouts are much longer than the reward model’s output. With asynchronous reward computation, using our grader as the reward signal therefore does not significantly hurt overall training speed. That said, we believe future work could work on distilling ProofGrader into a smaller reward model to further reduce the cost of RL training for math proofs. We emphasize that ProofGrader still serves as a necessary first step before such distillation becomes feasible.
> >
> > > **Q1. Since the best-of-n experiments are supposed to establish the predictive generalization power of ProofBench for use as a reward model, shouldn’t the ground truth annotations for the proofs used in the experiment be annotated by different people than those used to annotate the examples in ProofBench? A difficulty in designing a benchmark for evaluators is that the bias in the annotations themselves need to be accounted for, i.e, the paper needs to demonstrate that the possibly imperfect annotator preferences used in ProofBench are sufficient for generalization to general aesthetic preferences in the math community (regarding competition style proofs).**
> >
> > You are correct that if the best-of-n dataset contained similar biases to ProofBench, it would not adequately demonstrate generalization capability.
> >
> > As mentioned in the global response, our final annotation involves five experts with minimal overlap between datasets. ProofBench was primarily annotated by experts A, B, and C, while the best-of-n dataset was predominantly annotated by experts D and E. Expert B participated only in proofreading approximately 25% of the best-of-n proofs. Moreover, our consensus process works to minimize individual bias. In the best-of-n dataset, 50% of responses are double-graded, and disagreements are resolved through discussion between annotators. We tried our best to mitigate individual annotator biases and imperfect preferences.
> >
> > > **Q2. Similarly to Q1, in the best-of-n experiments, shouldn't the set of model generations being evaluated by the candidate evaluators be generated by models that are different to the three specific models used to generate the outputs in the benchmark? Could the authors provide justification why the current methodology is sufficient to demonstrate the general utility of ProofBench for selecting evaluators meant to evaluate generations from other models? Currently, it seems like ProofBench is useful for selecting evaluators for proofs generated by o3, Gemini, and DeepSeek-R1, but no indication of general ability.**
> >
> > Thank you for this question. The goal of the best-of-n experiments is not to test cross-generator generalization, but to evaluate whether our offline metrics (MAE/RMSE) reliably predict practical evaluator utility, and to demonstrate that fine-grained scoring outperforms binary scoring in downstream selection tasks. This purpose is distinct from the analyses in Section 3. Thus, using o3, one of the generators included in ProofBench and representative of frontier reasoning ability, is appropriate for this evaluation.
> > In realistic reward-modeling settings, evaluators are typically used to compare multiple candidate solutions from the same generator. Our best-of-n setup mirrors this usage: the central question is whether an evaluator can reliably rank the quality of multiple outputs across various problems for a fixed model, rather than whether it generalizes across unrelated generators.
> >
> > We agree that including additional generators could further strengthen the claim. As noted in our response to W1, we have already added results for weaker open-source models, providing complementary evidence that ProofBench’s insights extend beyond the three main generators. Expanding best-of-n experiments to many more generators would require substantial additional expert annotation effort, while adding limited value to our core conclusion.

---

> > > ### Author Response · Authors · 2025-11-20
> > > **Response to Reviewer HNs6 (3/3)**
> > >
> > > > **Q3. This approach seems very committed to a specific frozen mark scheme. Since the labor intensive annotation process is predicated on a particular MS, what happens if the mark scheme needs to be changed? Do we need to re-annotate each time? Could the authors provide justification why using a particular frozen mark scheme for the annotation process is sufficiently general to avoid re-annotation?**
> > >
> > > Thank you for raising this concern. First, we emphasize that a reference marking scheme is essential for human experts to grade proof solutions on a 0-7 scale consistently [1,2]. Without it, graders exhibit significant variance in assigning partial credit (i.e., determining what constitutes meaningful progress).
> > >
> > > We used a frozen marking scheme to ensure consistent application across all model generations, making it a reliable calibration tool. Regarding whether re-annotation is required if the marking scheme changes: yes, re-annotation would be necessary. This is the same for human math competitions: marking schemes are finalized before large-scale grading precisely because any changes would necessitate re-grading.
> > >
> > > In our experiments, we carefully designed the marking scheme to align with real competition standards as much as possible. If we ever decide to revise it, we can re-annotate using the same annotation interface, and the cost of that is still acceptable. The most labor-intensive part of annotation is understanding the semantic logic of a candidate’s proof. This understanding is preserved regardless of the scheme. Re-annotation would largely be a "re-mapping" exercise, primarily affecting partial credit distributions, rather than a restart. This requires substantially less effort than the initial pass.
> > >
> > > ---
> > > [1] Evan Chen. Guidance for problem captains (Or: how to write an olympiad rubric). https://web.evanchen.cc/static/usemo/captain-guidance-usemo.pdf.
> > >
> > > [2] Petrov et al. Proof or Bluff? Evaluating LLMs on 2025 USA Math Olympiad. https://arxiv.org/pdf/2503.21934.

---

> ### Author Response · Authors · 2025-11-26
> **Official Comment by Authors**
>
> Dear Reviewer HNs6,
>
> With the December 2 rebuttal deadline approaching, we wanted to reach out to see whether you have any follow-up questions or clarifications regarding our submission. We would be glad to address any remaining concerns and provide further explanation where helpful. If possible, we would appreciate receiving any comments at your earliest convenience.
>
> Thank you again for your time and thoughtful reviews.
>
> Best regards,
>
> The Authors

---

### Author Response · Authors · 2025-11-20
**Global Response (1/n)**

We thank all the reviewers for the valuable feedback! We are excited that many strengths of our paper are recognized by reviewers: our paper focuses on a very important, timely and impactful problem (HNs6, LGPu, zjS9, EDhE); the systematic methodology is well-motivated and well-designed (HNs6, LGPu, EDhE); the experimental design is comprehensive and thorough with clear ablation studies that provide actionable insights (HNs6, LGPu, EDhE); the best-of-n experiments effectively demonstrate practical utility (HNs6, LGPu,zjS9, EDhE); and ProofBench provides a valuable resource for the community as the largest expert-annotated dataset with continuous-scale grading (LGPu, zjS9, EDhE).


To address the reviewers' concerns, we have submitted a revised paper incorporating several changes, and shared our dataset as supplementary materials. Below, we first summarize the major revisions and address common questions raised by reviewers. We will provide individual responses to each reviewer's specific concerns.


## Revisions


We significantly improved the writing of the paper based on reviewers’ feedback, and respectfully ask the reviewers to take a look. Most concerns have been addressed in the revised version.


We have made the following updates to our dataset, with corresponding adjustments throughout the paper. **Importantly, none of the core claims, conclusions, or overall narrative of our work has changed.**
- Expanded ProofBench dataset: We have expanded ProofBench to 145 problems with 435 LLM-generated mathematical solutions, now covering all problems from the six competitions across the four-year period mentioned in the paper.
- Enhanced best-of-n experiments: We have expanded the best-of-n datasets to include 16 responses per problem (double the previous size) and implemented Monte Carlo subsampling for each n (replacing the previous approach of using only the first n responses). This enables evaluation across a larger range of n values and provides unbiased results. Figure 3 and descriptions in Section 4 have been updated to reflect these changes.
- Increased annotation coverage: We have added more expert graders to strengthen the dataset. The complete annotation process now involves 5 experts: in ProofBench, 41% of proofs are double-graded; in the best-of-n datasets, half of the responses (8 out of 16 per problem) are double-graded. Additional details are provided in Section 2.1 and Appendix A.3 of the revised paper.


We slightly adjust the paper structure for better presentation: section 2, 3, 4 now correspond to our three contributions respectively. The tables and figures are all improved for better presentation.

We provide our dataset artifacts in an anonymous repository [1].

## Common Questions

> **1. What are the inner-annotator agreement statistics? How do you resolve inconsistencies between annotators? (HNs6, LGPu, EDhE)**

As mentioned in the earlier paragraph, we have refined our dataset to strengthen annotation quality and coverage. The annotation process is now conducted by a team of five experts. In ProofBench, 41% of proofs are double-graded; in the best-of-n datasets, 50% of responses (8 out of 16 per problem) are double-graded. For double-graded data, we achieve within-1-point agreement (WTA≤1) 87.5%, which is higher than the agreement rates between AI graders and human experts reported in our paper.

To resolve inconsistencies, experts work collaboratively in supervised sessions monitored by a PhD student and a professor (both co-authors). When annotators differ by >1 point, they discuss their rationale to reach consensus. On the 0–7 scale, scores naturally cluster into four bands: incorrect (0), partial progress (1–3), nearly complete (4–6), and fully correct (7). During discussion, annotators must agree on (1) which band a solution belongs to, and (2) the exact score for fully correct (7) and fully incorrect (0) solutions. For partial-credit cases (1–6), minor differences may initially arise due to inherent ambiguities in natural-language proofs and subjective judgment even with marking scheme guidance, but these are resolved through further discussion. All scores in the final dataset reflect this consensus-based adjudication. Please find more details in section A.3 in the revised paper.

---
[1] Link to the datasets. https://anonymous.4open.science/r/ProofBench-Supplementary-4DD2.

---

> ### Author Response · Authors · 2025-11-20
> **Global Response (2/n)**
>
> > **2. Error analysis is absent. The paper does not systematically examine where ProofGrader fails or succeeds, what types of errors it makes (over-crediting vs. under-crediting), or which problem types are most challenging. (HNs6, zjS9)**
>
> We appreciate the concern about missing error analysis. After examining ProofGrader’s outputs, we find that cases of over-crediting and under-crediting by more than one point are relatively balanced (10.8% vs. 12.2%). Across generators, ProofGrader makes the most errors on o3-generated solutions–28.9% of the solutions, compared to 23.3% for DeepSeek-R1 and 16.9% for Gemini 2.5 Pro. Since ProofGrader itself is based on o3, this suggests a degree of within-generator bias. By problem type, ProofGrader fails more frequently on Algebra and Geometry problems (each accounting for ~25% of failure cases) than on other domains.
>
> For qualitative analysis, we manually examined 50 solutions for which two or more ensemble runs of ProofGrader deviated from expert scores by at least two points. This reveals several common failure modes:
>
> **Overcrediting**
> 1. **Appearance-of-completeness.** ProofGrader sometimes assigns high scores to solutions whose structure closely mirrors the rubric (sections, lemmas, final condition) even when the central mathematical claim is false, so superficial alignment masks a fatally incorrect core argument (e.g., o3 – APMO 2025/2).
> 2. **Fatal gaps are treated as minor omissions.** When a proof relies on a false universal claim or a broken reduction, ProofGrader often interprets this as “missing justification” rather than a fatal error, granting partial credit even though the entire method collapses (e.g., R1 – APMO 2024/3).
> 3. **Overtrust in sophisticated but incorrect frameworks.** For solutions that use polished frameworks (coordinates, height functions, custom transformations), ProofGrader tends to reward the global structure and technical vocabulary without verifying key validity conditions, leading to substantial overcredit for incomplete or invalid arguments (e.g., Gemini – APMO 2023/5).
>
> **Undercrediting**
>
> 1. **Penalizing early missteps despite a correct final proof.** LLMs often explore an incorrect approach before restarting with a valid proof; humans grade the final argument, but ProofGrader aggregates all attempts and heavily penalizes early errors, leading to near-zero scores even when the final proof is fully correct (e.g., R1 – APMO 2023/2).
> 2. **Double-penalizing a single flaw.** A single conceptual error can both prevent multiple checkpoints from firing and trigger a global “major error” deduction, so partially correct solutions with substantial progress can be driven all the way to 0/7 (e.g., R1 – Putnam 2022 B4).
> 3. **Overstrict demand for micro-justifications.** Omitted trivial steps or minor slips (e.g., a mislabeled point) can cause ProofGrader to mark long downstream chains as unproven, heavily undergrading solutions whose main ideas and overall argument would receive high partial credit from human graders (e.g., o3 – USAMO 2023/1).
>
> **Variance Category - Geometry Instability**
> - **Geometry dependency-chain instability (over- and undercrediting).** Geometry problems are particularly brittle: long dependency chains (similarity → ratios → concurrency/cyclicity) mean a single incorrect or missing step can either be wrongly ignored (leading to overcredit, e.g., treating a non–circle-preserving transformation as valid) or over-amplified (leading to undercredit when a routine angle equality is left implicit), producing highly unstable scores on geometric proofs (e.g., o3 – APMO 2024/1 vs. o3 – EGMO 2025/4).
>
> We have included these analyses in section 3.5 and Appendix A.8.

---

> ### Author Response · Authors · 2025-11-20
> **Global Response (3/3)**
>
> > **3. Computational costs and efficiency are not discussed. What are the computational costs of your best evaluator compared to simpler alternatives? (HNs6, LGPu)**
>
> This is a valid concern. We provide the cost analysis below:
> - **Latency:** With parallel or asynchronous processing, the overall latency is dominated by the slowest call, which can reach up to 5 minutes for strong reasoning models such as o3. In our ProofGrader (o3-based) experiments, the median request time is 75 seconds. Latency also varies significantly by provider, for example, when using OpenRouter for DeepSeek-R1, individual requests can take up to 10 minutes due to service instability and repeated retries.
> - **Financial cost:** We estimate financial cost using the number of prompt and completion tokens together with each provider’s official pricing. Below we report the cost of evaluators built on four different models across multiple context settings.
>
> | Evaluator Model | Context | Avg Input Tokens | Avg Output Tokens | Cost Range per Entry | Total Cost on ProofBench |   MAE |
> | --------------- | ------- | --------------- | ---------------- | -------------------- | ----------------------- | ----: |
> | ProofGrader     | Both    |                — |                 — | —                    |                    ~$116 | 0.926 |
> | o3              | Both    |             6805 |              4817 | \\$0.01– $0.09          |                   $23.19 | 0.964 |
> | o3              | MS      |             5076 |              4213 | \\$0.005–$0.19         |                   $19.47 | 1.069 |
> | o3              | Ref     |             5122 |              4645 | \\$0.003–$0.22         |                   $21.05 | 1.330 |
> | o3              | None    |             3654 |              6685 | \\$0.002–$0.29         |                   $26.99 | 1.680 |
> | Gemini-2.5-Pro  | Both    |             6900 |              9420 | \\$0.017–$0.239        |                   $45.66 | 1.342 |
> | Gemini-2.5-Pro  | MS      |             5067 |              8369 | \\$0.01–$0.218         |                   $39.97 | 1.142 |
> | Gemini-2.5-Pro  | Ref     |             5247 |             10977 | \\$0.009–$0.274        |                   $51.65 | 1.910 |
> | Gemini-2.5-Pro  | None    |             3701 |             12612 | \\$0.008–$0.131        |                   $58.05 | 2.107 |
> | o4-mini         | Both    |             6754 |              1697 | \\$0.004–$0.054        |                    $6.61 | 1.367 |
> | o4-mini         | MS      |             4985 |              1457 | \\$0.003–$0.048        |                    $5.28 | 1.234 |
> | o4-mini         | Ref     |             5121 |              1669 | \\$0.001–$0.052        |                    $5.76 | 1.504 |
> | o4-mini         | None    |             3654 |              1989 | \\$0.001–$0.049        |                    $5.67 | 1.914 |
> | DeepSeek-R1-0528     | Both    |             6370 |              6149 | \\$0.002–$0.072        |                   $12.64 | 1.357 |
> | DeepSeek-R1-0528      | MS      |             4720 |              5440 | \\$0.001–$0.108        |                   $11.25 | 1.298 |
> | DeepSeek-R1-0528      | Ref     |             4771 |              6569 | \\$0.003–$0.107        |                   $13.55 | 2.736 |
> | DeepSeek-R1-0528      | None    |             3425 |              8569 | \\$0.002–$0.129        |                   $17.04 | 2.842 |
>
> Our results show that:
> - Evaluators are actually cheaper when provided with contextual information, as they require less thinking—justifying our approach of incorporating richer contextual information.
> - Our best single-pass version is both inexpensive ($23.19 for the entire benchmark) and accurate (error rate of 0.964), and ensemble further lowers error rate at a higher cost.
> - Even the ensemble-based approach is far faster and significantly cheaper than expert grading.

---

### Author Response · Authors · 2025-12-03
**Summary of Discussion-Phase Updates and Score Changes**

Dear Chairs,

First of all, thank you for all your efforts in organizing ICLR!

We wanted to briefly summarize the outcomes of the discussion phase, as they are not reflected in the current scores. After we expanded ProofBench, released the dataset in an anonymous repository, added new analyses (error analysis, cost analysis, ensemble and ranking evaluations, weaker/open-source models, etc.), and clarified our annotation and rubric-generation pipeline, two reviewers explicitly updated their assessments, changing the overall scores from (6，6，4，4) to (8，6，6，4). **All score raises happened on November 21st.**

- Reviewer zjS9 **raised their score from 6 → 8**, noting that their concerns had been addressed.
- Reviewer EDhE **raised their score from 4 → 6**, explaining that most of their concerns were already resolved by the initial rebuttal; after our follow-up responses to additional questions, they stated that their remaining concerns were also sufficiently addressed and that they would **continue to update their score** during the discussion period.

For the remaining two reviewers, the discussion was cut off before they could reply, but our final responses **directly targeted their specific concerns**:
- Reviewer HNs6 focused on (i) ProofGrader’s generalization to weaker generators, (ii) inter-annotator reliability and error analysis, (iii) computational cost, and (iv) dependence on a fixed marking scheme. In response, we added experiments with weaker open-source generators (Qwen3-4B and DeepSeek-R1-0528-Qwen3-8B), substantially expanded our description of the annotation process and reliability, added a detailed error analysis with examples, provided a cost breakdown and justification to different evaluators, and clarified the rationale for fixing a marking scheme per problem while noting that supporting alternative schemes would require only modest re-annotation.
- Reviewer LGPu focused on (i) the size and scope of the dataset, (ii) potential bias from LLM-generated marking schemes, (iii) generalization beyond competition math, and (iv) comparison to other evaluation approaches. We expanded ProofBench to cover all problems from six major competitions over four years, described our multi-stage marking-scheme development and refinement process (with annotators treating the schemes as flexible guides), clarified that our claims are explicitly restricted to competition-style proofs while noting broader generalization as future work, and explained why simple heuristics or formal proof checkers are not directly applicable to natural-language competition proofs, positioning such methods as complementary rather than competing baselines.

We kindly ask that you consider these discussion-phase clarifications, the explicit score increases from zjS9 and EDhE, and our targeted responses to HNs6 and LGPu’s concerns when forming your recommendation.

Best regards,

The Authors

---

### Meta-Review · Area_Chair_Drua · 2026-01-07

**Summary:**

- **HNs6** (4): Questioned generalization to weaker models, inter-annotator reliability, error analysis, computational costs, and marking scheme rigidity. Flagged as Critical.
- **LGPu** (6): Concerned about dataset size (393→435 solutions), reliance on LLM-generated marking schemes, domain generalization, and lack of comparison with alternative evaluation methods.
- **zjS9** (6 -> 8): Initially noted need for case studies, MathArena comparison, and open-source model evaluation. Raised questions about self-enhancement bias and reward hacking.
- **EDhE** (4 -> 6): Critically flagged reproducibility (no dataset), insufficient reporting of human grading reliability, unclear rubric generation process, data contamination issues, and unclear realistic application scenarios.

**Reviewer Concerns:**

Most above issues are addressed by the authors.

**Reviewer Scores:**

zjS9: 6 -> 8
EDhE: 4 -> 6

---

### Decision · Program_Chairs · 2026-01-26

Accept (Poster)